# A study of optical scattering modelling for mixed phase Polar Stratospheric Clouds

Francesco Cairo[1], Terry Deshler[2], Luca Di Liberto[1], Andrea Scoccione[3], and Marcel Snels[1]

[1]Istituto di Scienze dell'Atmosfera e del Clima, Consiglio Nazionale delle Ricerche, Rome, Italy
[2]Department of Atmospheric Science, University of Wyoming, Laramie,Wyoming, USA
[3]Centro Operativo per la Meteorologia, Aeronautica Militare, Pomezia, Italy

**Correspondence:** Francesco Cairo (f.cairo@isac.cnr.it)

**Abstract.** Scattering codes are used to study the optical properties of Polar Stratospheric Clouds (PSC). Particle backscattering and depolarization coefficients can be computed with available scattering codes once the particle size distribution (PSD) is known and a suitable refractive index is assumed. However, PSCs often appear as external mixtures of Supercooled Ternary Solution (STS) droplets, solid Nitric Acid Trihydrate (NAT) and possibly ice particles, making questionable the assumption of a single refractive index and a single morphology to model the scatterers. Here we consider a set of fifteen coincident measurements of PSCs above McMurdo Station, Antarctica, by ground-based lidar and balloon-borne Optical Particle Counter (OPC), and in situ observations taken by a laser backscattersonde and OPC during four balloon stratospheric flights from Kiruna, Sweden. This unique dataset of microphysical and optical observations allows to test the performances of optical scattering models when both spherical and aspherical scatterers of different composition and, possibly, shapes are present.

We consider particles as STS if their radius is below a certain threshold value $R_{th}$ and NAT or possibly ice if above it. The refractive indices are assumed known from the literature. Mie scattering is used for the STS, assumed spherical, while scattering from NAT particles, considered as spheroids of different Aspect Ratio (AR), is treated with T-Matrix results where applicable, and of geometric-optics-integral-equation approach where the particle size parameter is too large to allow for a convergence of the T-matrix method. The parameters $R_{th}$ and AR of our model have been varied between 0.1 and 2 $\mu$m and between 0.3 and 3, respectively, and the calculated backscattering coefficient and depolarization were compared with the observed ones. The best agreement was found for $R_{th}$ between 0.5 and 0.8 $\mu$m, and for AR less than 0.55 and greater than 1.5. To further constrain the variability of AR within the identified intervals we have sought an agreement with the experimental data by varying AR on a case-by-case basis, and further optimizing the agreement by a proper choice of AR smaller than 0.55 and greater than 1.5, and $R_{th}$ within the interval 0.5 and 0.8 $\mu$m. The ARs identified in this way cluster around the values 0.5 and 2.5. The comparison of the calculations with the measurements is presented and discussed. The results of this work help to set limits to the variability of the dimensions and asphericity of PSC solid particles, within the limits of applicability of our model based on the T-matrix theory of scattering and on assumptions on a common particle shape in a PSD and a common threshold radius for all the PSDs.

# 1  Introduction

Polar Stratospheric Clouds (PSCs) appear in the polar stratosphere during winters due to the very low temperatures and the dynamic isolation of air within the polar stratospheric vortex. They are important because of the two roles they play in polar stratospheric ozone depletion: providing surfaces for the heterogeneous reactions that lead to the reactivation of chlorine and decreasing the concentration of $HNO_3$ in the gaseous phase, thus altering the balance of the chlorine activation/deactivation cycles (Solomon, 1988). A comprehensive review of studies and knowledge acquired can be found in Tritscher et al. (2021).

PSCs can either be formed of liquid droplets composed of supercooled ternary solutions (STS) of sulfuric acid, nitric acid and water, or solid nitric acid trihydrates (NAT), the thermodynamically stable form of $HNO_3$ and $H_2O$ in the polar stratosphere, or possibly - when temperature is low enough - ice. Initially PSCs were classified as three types based on lidar measurements of the intensity of the backscattered light and the amount of depolarization of the returned signals (Browell et al., 1990). With the accumulation of observations, it has been realized that it is not common to observe a PSC of a single well defined type (Pitts et al., 2018b). More often PSCs appear as external mixtures of liquid STS droplets, NAT, and/or ice, depending on the thermal history that led to their formation. For example, it is believed that the nucleation of NAT could start in droplets of a pre-existing population of STS, but not all liquid droplets may convert into solid NATs, allowing the coexistence of particles of different composition and phases and thus of intermediate optical characteristics (Peter and Grooß, 2012).

The existence of multi-phase PSCs with particles of different shapes and composition, hence different particle refractive indices, makes the modeling of the scattering characteristics of the cloud problematic. While Mie scattering theory has been used to analyse PSCs consisting of spherical particles (Toon et al., 2000), detailed analysis of observations of different classes of PSCs (Deshler et al., 2000) may be questionable because they may consist of non spherical solid particles with sizes comparable to the wavelength of the laser, and hence the results may be hampered by biases due to the unverified assumption of spherical scatteres. Because of this, some studies have chosen to limit themselves to liquid clouds only (Jumelet et al., 2008), or to make an effort and use theoretical modelling of light scattering for aspherical scatterers in the analysis. A viable solution which is not so demanding in terms of computational effort is the use of T-Matrix theory (see Liu and Mishchenko (2001), and references therein). T-Matrix method is an exact technique for the computation of aspherical scattering based on a direct solution of Maxwell's equations assuming homogeneous, rotationally symmetric non spherical particles or clusters of spheres (Mishchenko et al., 1996), and T-Matrix codes are orders of magnitude faster than other approaches used in particle light scattering, like the Discrete Dipole Approximation (Singham and Salzman, 1986) and the Finite Difference Time Domain (Yang and Liou, 1996a) techniques. The T-Matrix approach to compare microphysical and optical observations has been used in a number of cases (Voigt et al., 2003; Scarchilli et al., 2005), under the assumption of particles as prolate or oblate spheroids. However even this approach could be controversial given that, just as solid particles cannot be modeled as spheres, for similar reasons it is debatable that they can be modeled as spheroids, and biases can arise under that assumption as well. For instance, Reichardt et al. (2002) showed how under the hypothesis of spheroidal particle shapes, surface area density and volume density of leewave PSCs are systematically smaller by, respectively, 10–30% and 5–25% than the values found for mixtures of droplets, asymmetric polyhedra, and hexagons. Furthermore, there is no clear indication on what kind of aspect

ratio can be unequivocally assumed for the spheroidal case (Reichardt et al., 2004; Engel et al., 2013; Woiwode et al., 2016). Finally, T-Matrix codes suffer from convergence problems beyond a few dozen size parameters, especially for extreme aspect ratios (Mishchenko and Travis, 1998).

Given that there is still no completely satisfactory solution to tackle scattering from solid PSC particles due to the ambiguities that still persist about their shape, the use of Mie theory continues to be attractive and is used widely despite the unverified hypothesis of spherical scatterers. The speed of computation it offers is advantageous when used for statistical or climatological studies (Pitts et al., 2009, 2013), or when optical properties less dependent on the assumption of sphericity have to be calculated, as is in the case of extinction (David et al., 2012; Daerden et al., 2007). In PSC studies, Mie theory was also sometimes used to simulate aspheric particulate backscattering, employing corrections that take into account its reduction due to the asphericity of the scatterers (Snels et al., 2021; Cairo et al., 2022). However, the Mie theory cannot produce depolarized backscattering and attempts with simple empirical models to use it to mimic depolarization from aspherical particles do a very poor job in reproducing the measured depolarization (Cairo et al., 2022).

In our study we employ concomitant microphysical (i.e. Particle Size Distributions (PSD)) and optical (i.e particle backscattering and depolarization coefficients) measurements of PSCs when both liquid and solid particles are present and compare these with optical scattering computations done with codes capable of reproducing depolarization in backscattering. We consider particles as spherical STS if their radius is below a certain threshold value $R_{th}$ and NAT or possibly ice if above it, the respective refractive indices being known from the literature. Mie scattering is used for the spherical part of the PSD, while scattering from NAT particles, considered as spheroids of different Aspect Ratio (AR), is treated with T-Matrix results where applicable, and of geometric-optics-integral-equation approach where the particle size parameter is too large to allow for a convergence of the T-matrix method. The parameters $R_{th}$ and AR of our model have been varied between 0.1 and 2 $\mu$m and between 0.3 and 3, respectively, and the calculated backscattering coefficient and depolarization were compared with the observed ones in order to find the $R_{th}$ and AR providing the best agreement between the computation and the measurements.

The aims of this effort are both to verify the ability of the T-matrix approach to reproduce the observations from lidar/backscattersonde, once the PSDs are supposed known, and to provide a contribution to the estimation of the shape and size limits of the NAT PSC particles.

The question of the shape of NAT particles is in fact far from being clarified, and has important implications for the denitrification mechanisms of the polar stratosphere, an important step in the process that leads to the destruction of stratospheric ozone. In fact, the sedimentation of large NAT particles is considered one of the main causes of denitrification of the polar winter stratosphere (Di Liberto et al., 2015). Their settling time influences this process, and it is in turn dependent on the NAT particles shape and size, both of which determining their settling speed and lifetime, hence their denitrification efficacy. Woiwode et al. (2014) assumed significantly non-spherical NAT particles to simulate the NAT settling speed leading to a the vertical redistribution of $HNO_3$ observed between two companion flights during the RECONCILE airborne field campaign in the Arctic (von Hobe et al., 2013). Woiwode et al. (2016, 2019) have also suggested that NAT particles may be highly aspherical based on the infrared spectrometer MIPAS-STR limb observations exhibiting a spectral signature around 820 cm$^{-1}$ and an overall spectral pattern compatible with large highly aspherical NAT particles. T-Matrix calculations assuming randomly

oriented highly aspherical NAT particles (aspect ratios 0.1 or 10 for elongated or disk-like spheroids, respectively) were able to reproduce the MIPAS-STR observations to a large degree. Molleker et al. (2014) hypothesized strongly aspherical NAT particles to reconcile the amount of the condensed $HNO_3$ resulting from PSC cloud spectrometer measurements with the expected stratospheric values, and to provide consistency between particles settling velocities and growth times with back trajectories . Moreover, Grothe et al. (2006) observed highly aspherical NAT in laboratory experiments. This is in contrast with earlier studies that assumed an AR = 0.9 for the NAT spheroids to match microphysical model simulation with airborne (Carslaw et al., 1998) or satellite borne (Hoyle et al., 2013; Engel et al., 2013) lidar observations.

Finally, the methodology illustrated in the present work is not restricted to the study of mixed phase PSCs, but can find applications in all those cases in which the aerosol appears as an external mixture of solid and liquid particles, distinguishable on the basis of their different typical sizes.

## 2 Data and Methods

### 2.1 Dataset

We have analysed microphysical observations acquired by a balloon-borne OPC from fifteen Antarctic balloon flights, coincident with measurements of PSC backscattering and depolarization coefficients observed by a ground-based lidar. These observations were taken above McMurdo Station, Antarctica, between 1994 and 1999 (Snels et al., 2021). In addition we analysed four in situ balloonborne observations carried out by a laser backscattersonde and an OPC during 4 stratospheric balloon flights from Kiruna, Sweden, between 2000 and 2002 (Weisser et al., 2006). The Antarctic lidar and balloonborne OPC dataset has been extensively discussed in Snels et al. (2021) where it has been used to provide empirical relations linking particle Surface Area and Volume densities with the backscattering coefficients. The main characteristics of the instrumentation will be here only briefly recalled.

The lidar observations have been provided by a system detecting 532 nm backscattered light with parallel and perpendicular polarization with respect to the linear polarization of the emitting laser (Di Donfrancesco et al., 2000), thus allowing the measurement of Backscatter Ratio BR, Volume Depolarization $\delta$ and Aerosol Depolarization $\delta_A$ from 10 to 23 km. These optical parameters follows the usual definitions (Cairo et al., 1999). In the following the subscripts $mol$ and $A$ denote respectively the molecular and particle contribution to the optical coefficients, and $cross$ and $par$ denote the perpendicular and parallel polarization of the backscattering coefficient $\beta$ (Collis and Russell, 1976).

$$BR = \frac{\beta_A^{cross} + \beta_{mol}^{cross} + \beta_A^{par} + \beta_{mol}^{par}}{\beta_{mol}^{cross} + \beta_{mol}^{par}} \tag{1}$$

$$\delta = \frac{\beta_{mol}^{cross} + \beta_A^{cross}}{\beta_{mol}^{par} + \beta_A^{par}} \tag{2}$$

$$\delta_A = \frac{\beta_A^{cross}}{\beta_A^{par}} \tag{3}$$

$$\tag{4}$$

An alternative definition of Total Volume Depolarization $\delta_T$ and Total Aerosol Depolarization $\delta_{TA}$ will also be used in the following and is here introduced as:

$$\delta_T = \frac{\beta_A^{cross} + \beta_{mol}^{cross}}{\beta_A^{cross} + \beta_{mol}^{cross} + \beta_A^{par} + \beta_{mol}^{par}} \tag{5}$$

$$\delta_{TA} = \frac{\beta_A^{cross}}{\beta_A^{par} + \beta_A^{cross}} \tag{6}$$

$$\tag{7}$$

being:

$$\delta_T = \frac{\delta}{\delta + 1} \tag{8}$$

The formulas for switching from one to the other can be found in Cairo et al. (1999).

The BR is retrieved using the Klett algorithm where the attenuation correction follows Gobbi (1995). The $\delta$ is calibrated with the method described in Snels et al. (2009). Experimental errors in the particle Backscatter Ratio (R-1) are estimated to be 5%, but not less than 0.05 in absolute value, while the error in volume depolarization is about 10%–15%. Additional uncertainty comes from the determination of pressure and temperature from radiosoundings, needed to compute $\beta_A$ an $\delta_A$ (Adriani et al., 2004). Typical measurements are 30–60 min integration over the signal, and the vertical resolution is 75 m in 1994 and 1995 and 225 m in the other years. The OPC, which makes 10 second measurements, corresponding to roughly 50 meter vertical resolution, has been averaged to 250 meter bins. For comparison with the lidar the OPC has been interpolated onto the vertical grid of the lidar.

The OPC is described in Hofmann and Deshler (1991) and Deshler et al. (2003a). A through revision of its dataset is presented in Deshler et al. (2019). The instrument uses white light to measure scattering at $40°$ in the forward direction from particles passing through a dark field microscope. Mie theory and a model of the OPC response function are used to determine particle size throughout the range from 0.19 to 10.0 $\mu$m radius. The OPC provides time series of size resolved particle concentration histograms at 8 to 12 size bins, depending on the instrument used. A measurement of total concentration of particles is simultaneously determined by a condensation nuclei counter (CNC) close to the OPC, which grows all particles larger than 0.01 $\mu$m to an optically detectable size and counts them (Campbell and Deshler, 2014). Particle size histograms are fitted to unimodal or bimodal lognormal size distributions, which are the representation of size distribution used in this work. The uncertainties on the determination of the parameters of the mono/bimodal lognormals were determined by Deshler et al. (2003b) with Monte Carlo simulations. These were 20% for distribution width, 30% for median radii and 10% for modal particle concentrations.

A series of balloon launches were carried out from Kiruna, Sweden in the early 2000s. The payload included, among other instruments, and in addition to an OPC and a CNC, a backscattersonde capable of measuring in situ backscattering and depolarization at 532 nm with 10 seconds time resolution. Details of these instruments are presented in Adriani et al. (1999) and Buontempo et al. (2009). Here we use data from four flights that took place on 19 January 2000 (Voigt et al., 2003), 9 December 2001 (Deshler et al., 2003b), 4 and 6 December 2002 (Larsen et al., 2004; Weisser et al., 2006). As these balloon

flights were not simple ascents like the antarctic ones, but were commanded to perform altitude changes, by deflating the balloon or releasing ballast to maximize the transit time in the detected PSCs, the backscattersonde data have been interpolated to the OPC 250 m average of the data, corresponding to a time grid spacing of 60 or 75 seconds depending on flight.

In our study each data point includes the values of BR, $\delta_A$ and of the PSD defined by the three or six parameters of a mono or bimodal lognormal distribution. Altitude, pressure and temperature are also present as ancillary data. We identify a data

point as a PSC observation when the BR is greater than 1.2 and the temperature at the observation is below 200 K. Moreover, to select the presence of mixed phase clouds, we require that $\beta_A^{cross}$ is greater than $5\,10^{-6}km^{-1}sr^{-1}$. Under these conditions, a total of 141 data points from the Antarctic and 332 data point from the Arctic flights have been selected for the study.

## 2.2 Optical model

While the nucleation of NAT and ice is a threshold process, STS particles can grow upon cooling from the the ubiquitous

liquid Stratospheric Sulphate Aerosol (SSA), by continuously taking up nitric acid and water from the gas phase. They form droplets with volumes varying with temperatures but nevertheless with larger number density and smaller particle dimensions than NAT. Conversely, NAT particles are expected to be of smaller number density, but with dimension that can easily grow larger than the average STS particle radius of a few tenths of $\mu$m (Carslaw et al., 1997; Grooß et al., 2014) due to the smaller saturation vapour pressure of the nitric acid with respect to them. Deshler et al. (2003b) provide direct observations of this

separation between STS and NAT. All the more reason, due to the larger availability of water vapor, ice particles can grow even larger, often with linear dimensions exceeding 4–5 $\mu$m (Tritscher et al., 2019).

In our study we take advantage of the fact that in a mixed phase PSC the large particles are likely NAT or ice, solid particles which depolarize the backscattered light, while the small ones are liquid STS, that is spherical, and do not depolarize the backscattering. Figure 1 shows our dataset mapped in terms of the BR - as 1-1/BR for the sake of plot readability - and $\delta_A$ and

175 color coded with respect to the fraction of particles larger than 1 $\mu$m in radius, with respect to the total number of particles. As can be seen, it is a general feature that for each BR value, higher depolarization values are connected with higher fractions of large particles. Noteworthy, at high BR, high fractions of large particles are related with depolarizations which, albeit high, are smaller than those observed at medium-low BR. In fact those depolarizations at medium-low BR are associated with lower ratios of big to small particles. In other words, although at medium-low BR the large particles are proportionally less numerous

than at high BR, the depolarization is higher at medium-low BR. It has also to be noted in the plot that, in the high BR range at 1-1/BR=0.8, there are a few cases where high depolarization corresponds to a low fraction of large particles. In these cases, although the relative abundance of large particles was low, the particles were unusually large, exceeding a few $\mu$m. These very large particles, although in small concentrations, are causing the high depolarization observed. The temperature in these few cases was however high enough to exclude them as ice particles.

We plan to consider particles as STS when their radius is below a threshold value $R_{th}$ and NAT above it. We use values of 1.44, and 1.48 for the refractive index of, respectively, STS, and NAT. These values are compatible with the large PSC data set produced by the CALIPSO observations (Hoyle et al., 2013; Pitts et al., 2018a) and fall within the estimates presented for STS and NAT (Adriani et al., 1995; Deshler et al., 2000; Scarchilli et al., 2005). For completeness, ice particles are considered

when radii are larger than 4 $\mu$m and temperatures fall below 185 K. This happens only in 10% of the total dataset. In those few cases a value of 1.31 is used (Kokhanovsky, 2004).

For each data point, we split the PSD into two branches, namely $PSD_{STS} = PSD(r < R_{th})$ and $PSD_{asph} = PSD(r > R_{th})$. As stated, the presence of ice particles is taken into consideration by inspecting the temperature $T$ observed at the measurement, so if $T > T_{ICE}$ we pose $PSD_{NAT} = PSD_{asph}(r > R_{th})$, while if $T < T_{ICE}$ we limit the presence of NAT particles at radii smaller than $4\mu$m, i.e. $PSD_{NAT} = PSD_{asph}(R_{th} < r < 4\mu$m $)$ and consider as ice the particles with bigger radii, $PSD_{ICE} = PSD_{asph}(r > 4\mu$m$)$.

The backscattering coefficients and depolarization ratio for the STS, NAT and ice particles are separately computed. For STS we have used a Mie code (Bohren and Huffman, 2008), available from the NASA's OceanColor Web site. For NAT and ice we have used the GRASP (Generalized Retrieval of Aerosol and Surface Properties) Spheroid Package. GRASP is the first unified algorithm developed for characterizing atmospheric properties gathered from a variety of remote sensing observations (Dubovik et al., 2014), whose software packages are available on the Web. The Spheroid Package allows fast, fairly accurate, and flexible modeling of single scattering properties of randomly oriented spheroids with different size and shape. It includes a software and kernels data base. The details of the scientific concept are described in Dubovik et al. (2006). The kernel look-up tables include results of calculations using T-Matrix code where convergence was acquired, and when convergence was not achieved, the geometric-optics-integral-equation approach (Yang and Liou, 1996b, 2006) was used, that is expected to provide accurate optical characteristics for spheroids with size parameter larger than 30 – 40. The two methods have been shown to produce comparable results over the size range in which both are applicable (Dubovik et al., 2006). Thus, the software and kernels data base provide the kernel matrices for randomly oriented spheroids with Aspect Ratios (AR) from $\sim 0.3$ (flattened spheroids) to $\sim 3.0$ (elongated spheroids) and covering the size parameter range from $\sim 0.012$ to $\sim 625$ (when a wavelength of 0.44 is used) for a wide range of particle refractive index.

The total particle backscattering from the particles of the PSD can thus be parametrized with $R_{th}$ and AR and written as:

$$\beta_A = \beta_A(R_{th}; AR) = \beta_{STS}(R_{th}) + \beta_{NAT}(R_{th}; AR) + \beta_{ICE}(R_{th}; AR) \tag{9}$$

Here we have used for the sake of simplicity the same AR for NAT and ice. Even if this hypothesis is not fully verified, this should not impact severely our study, as only 10% of our observations have temperatures below 185 K, and no definite ice observations could be clearly discerned in our database. Now, the particle depolarization $\delta_A$ can be parametrized in terms of AR and $R_{th}$ as well and be written as a weighted average of the contributions from the different classes of particles:

$$\delta_A = \delta_A(R_{th}; AR) = \frac{\beta_{NAT}(R_{th}; AR) \cdot \delta_{A,NAT}(R_{th}; AR) + \beta_{ICE}(R_{th}; AR) \cdot \delta_{A,ICE}(R_{th}; AR)}{\beta_A(R_{th}; AR)} \tag{10}$$

It is useful here to recap how scattering from aspheric particles changes, compared to Mie theory, when T-Matrix is used. Obviously, Mie theory cannot reproduce the depolarized backscattering typical of aspherical scatterers. According to T-Matrix, the depolarization is negligible for scatterers size parameters lower than unity (given the wavelength used in our study, this corresponds to particle radius approximately below 0.1 $\mu$m), it grows up to a maximum that is reached for size parameters of the order of ten (i.e. particle radius around 1 $\mu$m) and then decreases towards an asymptotic value for size parameters greater

than one hundred (particle radius greater than 10 $\mu$m). Both the maximum value and the asymptotic value vary according to the AR considered. In particular, the asymptotic value of the depolarization can assume values from 10% to 40%, and the maximum value from 30% to 80%, the two variabilities are not connected. The dependence of the single particle depolarization on shape and size has been studied extensively by Liu and Mishchenko (2001). It has to be stressed that there is no simple relationship that binds the peak and asymptotic depolarization values to the AR of the particle, although there is a general tendency for small AR values to give large asymptotic depolarization values, and for large AR values to produce medium asymptotic depolarization values. The backscattering itself reproduces the Mie results for size parameters below unity, then is progressively reduced to values that can even be one third of the Mie value when the particle size parameter is above ten. Again this reduction depends in no simple way on the value of AR.

It is quite possible that in some case spheroids are not able to fully replicate the scattering properties of PSC particles. Um and McFarquhar (2011) used geometric ray-tracing codes on ice particles of linear dimension of few tens of $\mu$m and showed that differences in the backward scattering between different shape models (Chebyshev particles, Gaussian random spheres, and droxtals) are higher than 100%. Furthermore, it is possible that the assumption of the same AR for each particle in the solid part of the PSD, irrespective of its composition or size, can not hold; on the contrary it is quite possible that particle shape changes with size due to the differential condensation growth along preferred dimensions. Laboratory studies (Grothe et al., 2006), have shown that synthesized aspherical NAT develops different morphologies depending on growth conditions. However, our approach maintains an AR common to all PSD particles, to be considered as the average AR of the PSD. This is a simplistic choice, but in contrast we are not able to justify a particle size dependent AR on physical grounds. An additional uncertainty on the measured particle size is that aspherical particles in the OPC will scatter differently than the spherical particles assumed in the OPC retrieval.

## 2.3 Variability with the threshold radius $R_{th}$ and Aspect Ratio AR

We have computed $\beta_A(R_{th}; AR)$ and $\delta_A(R_{th}; AR)$ for a set of threshold radii $R_{th}$ and ARs ranging respectively from 0.1 to 2 $\mu$m and from 0.3 to 3. To find the values that provide the best match with the measured ones $\beta_A^{meas}$ and $\delta_A^{meas}$, we have calculated the respective Root Mean Squared Errors (RMSE), as:

$$RMSE_{\beta_A} = \sqrt{\frac{\sum_{i=1}^{n}(\beta_{Ai} - \beta_{Ai}^{meas})^2}{n}} \tag{11}$$

$$RMSE_{\delta_A} = \sqrt{\frac{\sum_{i=1}^{n}(\delta_{Ai} - \delta_{Ai}^{meas})^2}{n}} \tag{12}$$

where the index $i$ runs over our dataset. Covariance has also been computed resulting to be everywhere close to zero, except for $R_{th}$ smaller than 0.5 $\mu$m, where it resulted positive for AR between 0.55 and 1.5 (and close to unity for AR=1.25), while it was slightly negative (correlation between -0.3 and 0) for AR smaller than 0.55 and greater than 1.5. Since the range of variability of $\beta_A$ is two orders of magnitude, to estimate the goodness of the agreement independently of the magnitude of $\beta_A$, we have

also calculated the Root Mean Squared relative Error (ReRMSE), defined as:

$$ReRMSE_{\beta_A} = \sqrt{\frac{1}{n}\sum_{i=1}^{n}\left(\frac{\beta_{Ai}-\beta_{Ai}^{meas}}{\beta_{Ai}^{meas}}\right)^2} \tag{13}$$

$$ReRMSE_{\delta_A} = \sqrt{\frac{1}{n}\sum_{i=1}^{n}\left(\frac{\delta_{Ai}-\delta_{Ai}^{meas}}{\delta_{Ai}^{meas}}\right)^2} \tag{14}$$

The color coded RMSEs, (upper panel) and ReRMSEs (lower panel), with respect to $R_{th}$ and AR are shown in Figures 2 for backscattering and 3 for depolarization.

The plots reported in Figures 2 and 3 suggest how to select $R_{th}$ and AR to provide the best match between computations and observations. The comparison between the RMSE and ReRMSE for $\beta_A$ shows similar features: both plots suggest to avoid AR values between 0.6 and 1.5 when an $R_{th}$ below 1 $\mu$m is considered, and show similar minimum differences between model
and observations for AR below 0.55 and above 1.5. In this range of variability for AR, the $R_{th}$s that reach the best agreement between model and observations are between 0.3 and 1 $\mu$m, with 0.5-0.7 $\mu$m the most favorable. The analysis of the RMSE and ReRMSE for $\delta_A$ clearly shows regions where the agreement model-experiment is really poor, namely for AR below 0.75 and above 1.25 when $R_{th}$ is below 0.75 $\mu$m. Noteworthy, these regions only partially coincide with those of disagreement for $\beta_A$. The relative error is considerably higher than in the case of the comparison of $\beta_A$. Regions in which the agreement seems
better are those with $R_{th}$ greater than 0.5 $\mu$m and AR values below 0.75 or greater than 1.25. The result of this study allows to identify only the best $R_{th}$, resulting around 0.5-0.8 $\mu$m, while the ARs compatible with the measurements are all those between 0.3-0.55 and 1.5-3.

To further constrain AR we have kept $R_{th}$ at a fixed value, chosen between 0.5 and 0.8 $\mu$m and changed this value with a 0.1 $\mu$m step. For each of these fixed $R_{th}$, and separately for each PSD, we identified in the intervals (0.3-0.55), (1.5,3) the value of
AR which best matched the observed $\delta$ with its computed value. Finally, for each PSD we selected the pair $R_{th}$ and AR which provided the best match. Once the ARs and $R_{th}$ have been selected by forcing the agreement between the $\delta_A$s, the same ones have been used for the calculation of the $\beta_A$s.

## 3    Results

Figure 4 reports the scatterplot of measured vs computed $\beta_A$, colour coded in terms of AR. The Figure represents the analogue
of Figure 4 in Snels et al. (2021), where in the present case we have used a larger dataset, including now four Arctic balloon flights, and used T-Matrix instead of a factor 0.5 reduction in the Mie backscattering. Figure 5 reports the scatterplot of measured vs computed $\delta_A$ similarly color coded in terms of AR. The uncertainties associated with the measured $\beta_A$ and $\delta_A$ derive from the error analysis for the single lidar data, which can be found in Adriani et al. (2004) or from the standard deviation for the averaged data, depending on which is greater. The uncertainties on the calculated $\beta_A$ and $\delta_A$, are 40% as determined
by Deshler et al. (2003a) for any moment of a PSD derived from the OPC measurements. Deshler et al. determined this

through a Monte Carlo simulation which used the uncertainties of the OPC size and concentration measurements to quantify the uncertainties in the PSD parameters and their subsequent moments.

Despite the dispersion in Figure 4 the points cluster around the straight line $\beta^{calc}=\beta^{meas}$, indicating the agreement between computation and measurements can be considered fine for $\beta_A$ with the exception of $\beta$ values below $4 \cdot 10^{-5}km^{-1}sr^{-1}$ where the $\beta^{calc}$ underestimate the measurements. Such underestimation seems to be of the order of $10^{-5}km^{-1}sr^{-1}$, of the same order as the backscattering from the background atmospheric particulate matter in volcanic quiescent conditions, a magnitude compatible with possible inaccuracies in the calibration of the lidar data. The Pearson correlation coefficient for the entire dataset is 0.56, and increases if the lower values of $\beta$ are neglected.

The $\delta_A$ scatterplot shows the presence of a good number of points that align along the $\delta^{calc}=\delta^{meas}$ correlation line, with AR selected mainly around the value 0.5. However, for depolarization values greater than 30% there is no AR that will reproduce the measurements. These points correspond to those presented in Figure 1, with low values of BR and high values of the concentration ratio of large to total particles. They mainly come from three single observational periods of about one minute each, characterized by air temperatures between 184-188 K. Given the magnitude of the depolarization, it is possible that those observations are not referable to clouds in mixed phase, but rather to clouds of predominantly solid particles. For that particular set of points, we also explored the possibility that all particles were solid, but even under this assumption the comparison with the experimental data did not improve appreciably.

In Figure 5 for depolarizations lower than 15%, the points which deviate, by excess or defect, from the 1:1 straight line have predominantly AR greater than 1.5. So it seems that selected ARs greater than 1.5 generally produce a worse correlation. From Figure 4 we observe that AR values in the range (0.3-0.55) tend to be associated with medium-low $\beta$ values, while AR values in the range (1.5-3) are mainly associated with medium-high $\beta$.

To conclude, the choice of $R_{th}$ in a range between 0.5 and 0.8 $\mu$m leads to a reasonably good agreement between the $\beta$'s, but there seems to be a discrepancy between the calculated value and the measurements in their lower range of variability. From Figure 4 such mismatch, which makes the measurements larger than the calculations, seems to be of the order of $10^{-5}km^{-1}sr^{-1}$. The selection of the AR that produces the best agreement with the observed $\delta$'s leads to three results: i. The ARs in the range 0.3-0.55 tend to be selected in correspondence with medium-low $\beta$'s, the ARs in the range 1.5-3 in correspondence with medium-high $\beta$'s. ii. ARs in the 0.3-0-5 range reproduce the measurements well, except for some observations where the depolarizations are greater than 30%; iii. the ARs in the 1.5-3 range reproduce the measurements less well; iv. There is no AR that will allow the calculations to reproduce the measurements for depolarizations greater than 30%.

## 4   Discussion

The identification of the best $R_{th}$ in the range 0.5 -0.8 $\mu$m supports what we already know from the theoretical understanding of NAT particle formation in PSC and from measurements (Deshler et al., 2003b). Concerning particle shape, in our model all solid particles in a single PSD share the same AR, but different PSDs can have different ARs. This approach could suggest that

the choice of the AR which, case by case, optimizes the agreement between calculations and measurements, may be the result of chance rather than physics. There are two facts that counter this criticism.

First, it appears that the selected ARs may be related to the shape of the PSD. Figure 6 shows the 2D-histogram by occurrence of ARs and of $N(r > 0.7\mu m)/N_{tot}$, the ratio between particles with radius greater than 0.7 $\mu$m and total particles, which is a parameter related to the PSD shape. In Figure 6 the AR are not distributed randomly. Conversely, there is a tendency for the AR to grow as the percentage of large particles increases. In fact AR values tend to peak around 0.5 in the lower $N(r > 0.7\mu m)/N_{tot}$ range, while tend to cluster around 2.5 when $N(r > 0.7\mu m)/N_{tot}$ is higher. The shape of the PSD mirrors particle formation conditions and history, is linked to the presence of solid particles, as already highlighted in the discussion of Figure 1, and is likely linked to the average particle shape as well.

Second, if we consider the sequences of measurements acquired in individual balloon flights, the corresponding sequences of selected ARs do not evolve randomly but, conversely, are auto-correlated. An example of this behavior is shown in Figure 7, where the time series of $\beta$ and $\delta$ are reported respectively with red and blue dots. The ARs that provide the best agreement between experiment and simulation are shown with black dots. It can be seen that temporally contiguous observations often result in the selection of the same AR. Contiguous observations of PSD are likely to have similar characteristics in terms of microphysics, and this seems to be correctly reflected in the constancy of AR. We are therefore confident that our method produces results with a physics-based content.

In general our model leads to good correlations between measured and modeled $\beta$s. For the $\delta$s the measurements are well reproduced by the calculations in many instances, as is the case for many of the selected ARs in the range 0.3-0.55. However, there are other cases in which the agreement is worse (when the best ARs have been selected in the range 1.5-3), or does not occur at all, as in the cases of observed depolarizations greater than 30%. In these latter cases, the impossibility of reproducing the observed values even under the hypothesis of a completely solid particles implies that, for those PSDs, our model is not able to produce the observed depolarizations. In these particular cases in which the model performs particularly badly, there may be problems of inhomogeneities of the cloud. These cases come from Antarctic observations, for which the microphysical obser-vations from the balloon and the optical ones from ground-based lidar are separated geometrically, so that the two instruments sample air masses separated by several tens of kilometres, and it may be the case that some clouds were not homogeneous on such spatial scales.

Different shapes produce different polarization, according to T-Matrix. This has also been proven experimentally since the early work of Sassen and Hsueh (1998) and Freudenthaler et al. (1996) that showed how lidar depolarization ratios in persisting contrails ranged from 10% to 70%, depending on the stage of their growth and on temperature. In the T-matrix theory, for fixed AR, the depolarization depends on the particle size and maximizes for particular sizes. There is certainly a way to assume a particle size-dependent AR in our PSDs so as to reconcile the computations with the observed values. However such an approach would have little physical basis and could only be justified to maximize the agreement of calculations. Therefore we have not explored this possibility further, although it is possible that our simplified hypothesis of a common AR for every particle may be the cause of the bad agreement between data and calculations in some case.

To further investigate the causes of the mismatch we turn to the study of the climatology of PSC observations collected from McMurdo's lidar. The measurement of a PSC composed exclusively of solid particles is a rare and uncertain event. The absence of liquid aerosols is difficult to determine for certain. However, Adachi et al. (2001) demonstrated that in a plot of the Total Volume Depolarization $\delta_T$ versus 1-1/BR, the experimental points of solid, liquid or variously mixed PSCs are distributed within a triangle whose vertices are (1, 0), (1, $\delta_{TA}^{asph}$) and (0, $\delta_{mol}$). These vertexes represent respectively the value of $\delta_T$ in the case of pure liquid clouds and pure solid clouds for $BR = \infty$, when the $\delta_T$ coincides with $\delta_{TA}$, and in the case when no particles are present the $\delta_T$ attains its molecular value $\delta_{mol}$ (Young, 1980). Hence the extrapolated intercept on the y axis at $BR = \infty$ is precisely $\delta_{TA}^{asph}$. This procedure allows us to estimate this asymptotic value. This requires the assumption that the experimental points that fill the triangle of vertices defined above represent PSC observations in mixed phase in which all solid particles share the same aerosol depolarization. Alternatively, one can interpret differently the presence of the data points filling the triangle. These points may as well represent single phase PSC of solid particles but with different shapes, hence producing various depolarizations.

Figure 8 reports a 2D-histogram of $\delta_{TA}$ towards 1-1/BR from twelwe years of lidar observations from 1990 to 2002 in the antarctic station of McMurdo (Adriani et al., 2004). Despite the dispersion of the experimental points, a value close to 40%, as the highest vertex of the triangle, on the 1-1/BR=1 axis, for $\delta_{TA}^{asph}$ can be assumed. The corresponding value for $\delta_A$ is close to 70% according to eqns. (3), (5) and (6).

If we assume that the difficulty of our model to reproduce the observed depolarization in some case is due to the incorrect assumption of a common AR for all solid particles, we are led to interpret Figure 8 admitting that the experimental points filling the triangle of vertices (1, 0), (1, $\delta_{TA}^{asph}$) and (0, $\delta_{mol}$) represent both PSC in various degrees of mixed phase, and PSC in purely solid phase but composed of particles of various shapes. These various shapes give rise to different $\delta_{TA}$ between 0 and $\delta_{TA}^{asph}$ at the vertex of the triangle.

## 5 Conclusions

We have used an optical model to compute with T-Matrix theory the backscattering and depolarization of mixed phase PSCs. The model assumes that: i. PSC particles are solid (NAT or possibly ice) above a threshold radius $R_{th}$, liquid (STS) below; ii. A single shape is common to all solid particles in a PSD, irrespective of their size or composition. We have tested the model using a data set of coincident lidar, backscattersonde and OPC measurements from Antarctica and Arctic balloon flights.

While the agreement between modeled and measured backscattering coefficient is generally reasonable, there are cases in which it is less so for depolarization. The most likely reason is our simplified hypothesis of a common shape for all the solid particles present in the size distribution. However, our analysis has provided the range of optimal $R_{th}$ and AR parameters that best match the observations. To sum up: i. in an externally mixed PSC, it is reasonable to place a threshold radius $R_{th}$ between 0.5 and 0.8 $\mu$m, which divides the liquid part from the solid part of the particulate; ii. It is sensible to expect strongly aspherical shapes for the solid part of the cloud; iii. There are cases, in particular those related to high depolarization observations, which,

within our assumptions (i.e. a single form for the solid particles, a fixed threshold radius for all PSDs) precludes reproducing

the observed depolarization with a T-matrix approach.

*Code and data availability.*  The Mcmurdo lidar data are available at the NDACC web site

ftp://ftp.cpc.ncep.noaa.gov/ndacc/station/mcmurdo/ames/lidar/.

The OPC data files and size distributions are reported at the web site hosting the Wyoming in situ data

http://www-das.uwyo.edu/~deshler/Data/Aer_Meas_Wy_read_me.htm

and can be downloaded from

ftp://cat.uwyo.edu/pub/permanent/balloon/Aerosol_InSitu_Meas/Ant_McMur.

The arctic balloonborne backscattersonde data are available from the author upon request.

The software for Mie computation is available at:

https://oceancolor.gsfc.nasa.gov/docs/ocssw/bhmie_8py_source.html

The Spheroid Package of the software GRASP is available at:

https://www.grasp-open.com/products/spheroid-package-release/

*Author contributions.*  FC was responsible for most of the writing, review and editing process, supported by all co-authors. FC and MS share the idea behind the article and the data analysis work. MB helped in software development. TD provided OPC data and PSD analysis. LDL and AS provided for the identification and quality check of the dataset.

*Competing interests.*  No competing interests are present

*Acknowledgements.*  The authors acknowledge the financial support by PNRA in the framework of the projects POAS (Particles and Ozone in the Stratosphere of Antarctica) and ACLIM (Antarctic Clouds Investigation by Multi-instrument measurements and modeling). The OPC measurements were supported by awards from the US National Science Foundation (NSF) which include OPP award numbers 9615198, 9980594, and Arctic sciences award 0095158. Terry Deshler and Luca Di Liberto acknowledge a grant from the Short-Time-Mobility

program of CNR, respectively in 2016 and 2009. The arctic balloon flights were supported by the Commission of the European Union through the Environment and Climate Program (contract ENV4-CT97-0523) and through the CIPA program (EVK2-CT-2000-00095).

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

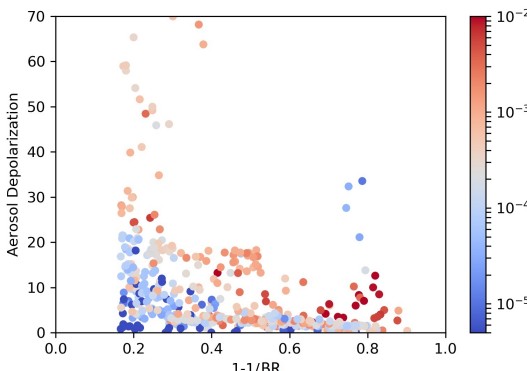

**Figure 1.** Scatterplot of aerosol depolarization vs 1-1/BR, where BR is the Backscatter Ratio. Data are from McMurdo lidar and Kiruna balloon flights backscattersonde, coincident with balloonborne OPC PSD measurements, fitted with mono or bimodal lognormals. The color codes the fraction of particles with radius larger than 1 $\mu$m with respect to the total number of particles in the PSDs. We report data points with BR greater than 1.2, $\beta_A^{cross}$ greater than 5 $10^{-6}km^{-1}sr^{-1}$ and temperature at the observation below 200 K.

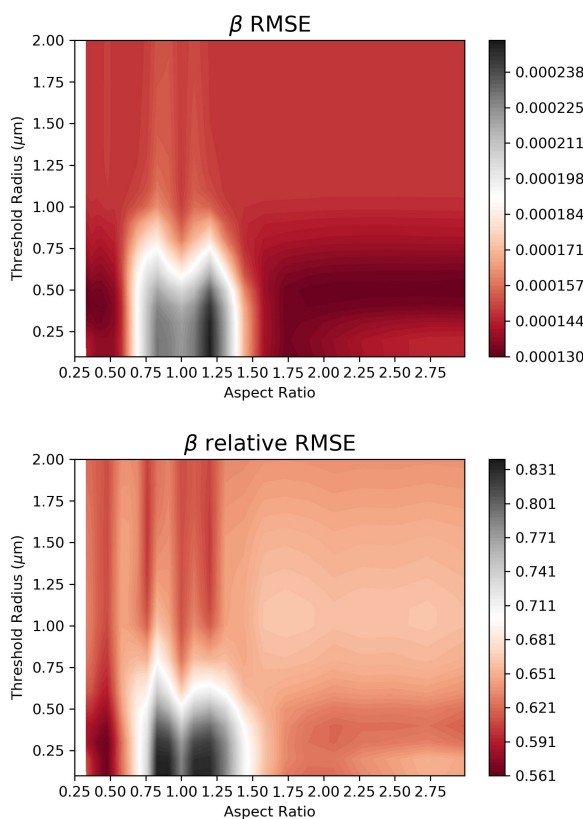

**Figure 2.** Contour plot of the RMSE (upper panel) and relative RMSE (lower panel) of the measured and modelled aerosol backscatter coefficient $\beta_A$.

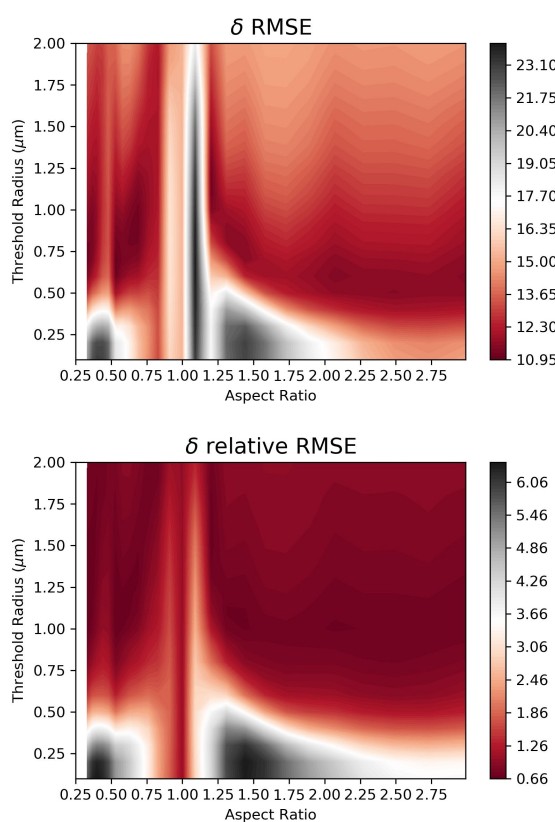

**Figure 3.** Contour plot of the RMSE (upper panel) and relative RMSE (lower panel) of the measured and modelled aerosol depolarization $\delta_A$.

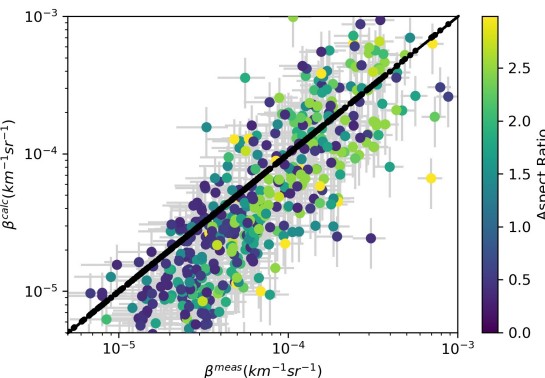

**Figure 4.** Scatterplot of computed vs measured particle backscattering coefficients $\beta_A$. The ARs used for the $\beta$ computations have been selected, case by case, to produce the best agreement between the $\delta$ computed and measured, and are here represented in color coding. Only ARs in the intervals between 0.3 and 0.55, and between 1.5 and 3, have been considered. $R_{th}$ was also selected within the interval 0.5-0.8 $\mu$m to provide the best match. We report data points with BR greater than 1.2, $\beta_A^{cross}$ greater than 5 $10^{-6}km^{-1}sr^{-1}$ and temperature at the observation below 200 K.

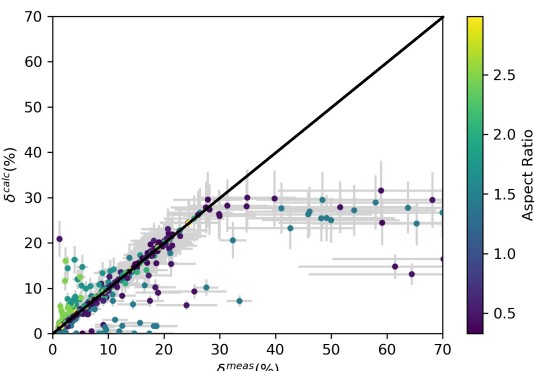

**Figure 5.** Scatterplot of computed vs measured particle depolarization $\delta_A$. The ARs used for the computations are those that provided the best match between the $\delta$ computed and measured, and are here represented in color coding. Only ARs in the intervals between 0.3 and 0.55, and between 1.5 and 3, have been considered. $R_{th}$ was also selected within the interval 0.5-0.8 $\mu$m to provide the best match. We report data points with BR greater than 1.2, $\beta_A^{cross}$ greater than 5 $10^{-6}km^{-1}sr^{-1}$ and temperature at the observation below 200 K.

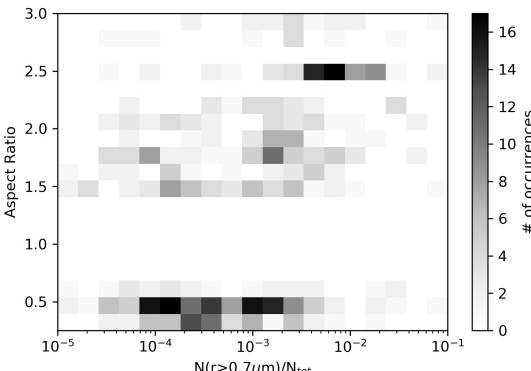

**Figure 6.** 2D-histogram of occurrence of ARs and of $N(r > 0.7\mu m)/N_{tot}$, the ratio between particles with radius greater than 0.7 $\mu$m and total particles. Only ARs in the intervals between 0.3 and 0.55, and between 1.5 and 3, have been considered.

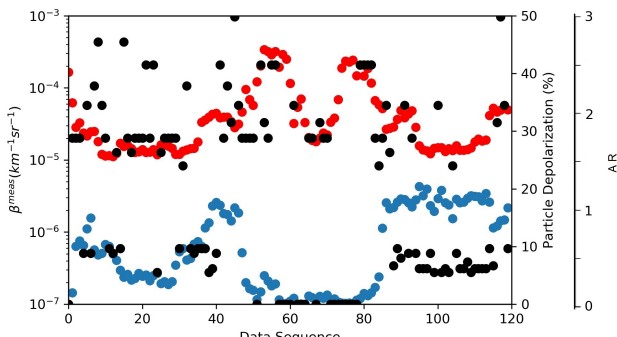

**Figure 7.** Sequences of $\beta$ (red dots) and $\delta$ (blue dots) measured on a balloon flight on December 9th 2001, from Kiruna, Sweden. Each data point represents an average over 60s. Black dots represents the ARs providing the best match between the $\delta$ and those computed from concomitant measurements of PSD.

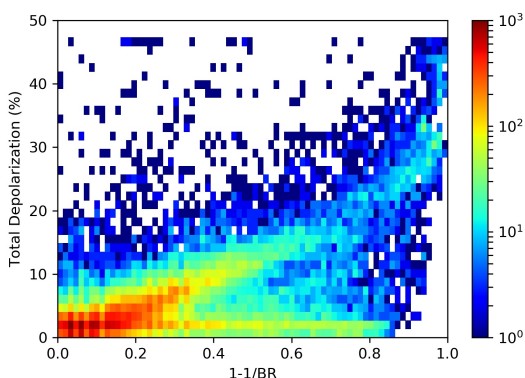

**Figure 8.** 2D histogram of Total Volume Depolarization $\delta_{TA}$ vs 1-1/BR, where BR is the Backscatter Ratio. Data are from McMurdo lidar and cover the winters from 1990 to 2002. The color codes the number of observations.