# Peer review of "A study of optical scattering modelling for mixed phase Polar Stratospheric Clouds"

_EGUsphere, 2022_

## Referee Comment (RC1)

Manuscript Review
Cairo et al.

The study that is described in this manuscript purports to evaluate methods for computing light scattering from aspherical particles using the T-matrix technique. The approach is to use PSDs measured in situ with an OPC and to calculate backscattering coefficients and depolarization ratios that are then compared to backscattering and depolarization measured directly with remote sensing techniques. It is my opinion that this manuscript was submitted prematurely and I recommend that the authors withdraw this submission and resubmit once they have addressed a number of issues that I list below. My decision to reject this paper is because the issues that I raise will require more than just a major revision and the paper has little to offer in its present form that can provide the scientific community with information that is useful in understanding PSCs or that can assist in interpreting measurements from in situ or remote sensors.

In their introduction, the authors state: *"The aim of our study is to employ concomitant microphysical and optical measurements of PSCs when both liquid and solid particles are present and compare these with optical scattering computations done with codes capable of reproducing depolarization in backscattering. This allows us to verify the capacities and limits of these codes, tested on a relevant data set which provides both microphysical and optical measurements of mixed-phase particulate distributions. The methodology illustrated is not restricted to the study of mixed phase PSCs, but can find applications in all those cases in which the aerosol appears as an external mixture of solid and liquid particles, distinguishable on the basis of their different typical sizes."*

By their own admission, they were unable to verify the capacities and limits of the code, not because of any flaws in the code itself, but most like due to how they applied it under very simplistic assumptions. I would add that I think that how they applied the code was also too simplistic, e.g. assuming that all the particles had the same AR rather than trying combinations of size-dependent AR.

Again, in my opinion, the concluding statement that *"However, the result for the backscatter coefficient simulation allows to constrain the model parameters Rth and AR into reasonable ranges and should be regarded as the positive result of the study"* does not justify publishing this study in its current form. The reason for this opinion is because the authors never explain how their study will benefit the science, not in the introduction nor in the concluding remarks; hence, the questions are 1) What purpose does constraining the model parameters serve?, 2) On what basis do they declare that these ranges are reasonable? and 3) Since they conclude that using prolate and oblate spheroids to model the scattering did not lead to useful results, that constraining the AR range is meaningless.

I think that for this study to be published and provide useful information to the broader scientific community, it must do the following:

1) Explain the importance of knowing the particle sizes and shapes in mixed phase PSCs.
2) Diagnose their model results to understand why they are producing unacceptable comparisons.
3) Run a sensitivity analysis using simulated PSCs and measurements to quantify the observed difference.
4) Correct a large number of typographical and grammatical errors that made the current manuscript distractive to read.

Other comments, questions and suggestions:

1) From the abstract onward the authors erroneously talk about comparing "microphysical and optical" measurements. This makes no sense since all of the measurements are microphysical and optical, i.e. the OPC uses an optical technique to derive size distributions that help describe the microphysical properties of the PSCs. Likewise, the remote sensing techniques are optical and are also used to derive microphysical properties of PSCs.
2) The authors never explain the relative importance of mixture of particle types in mixed phase PSCs. Had the modeling exercise been successful, who would benefit?
3) Nothing is discussed about the contribution to the backscattering of other types of stratospheric particles, e.g., meteoritic dust, sulfate particles, etc. How does that impact the measurements and modeling?
4) The backscatter instrument described by Adriani (1999) had multiple wavelengths. Why is only the 532 being used? Wouldn't modeling multiple wavelengths have improved the retrievals?
5) If this was a true modeling study, an iterative methodology should have been used to vary the mixtures of shapes and sizes until most closely matched by the measurements.
6) How homogeneous are these clouds and what do the PSDs look like derived from the OPC? The reader never sees the actual shapes of PSD or what the number concentrations are. This is important because it will impact the backscattering and depolarization. It is stated in the results section that apparently the larger particles are biasing the depolarization but this depends on the total concentration of particles and how homogeneous the mixture is. I could not find in the Adriani (1999) paper what beam volume is at each measurement gate.
7) In Figs. 4 and 5, there is no noticeable difference between AR-0.5 and AR=1.5. This does not surprise me because if you have an ensemble of randomly oriented spheroid, an oblate spheroid will look like a prolate spheroid, depending on their relative orientations; hence why even use ARs < 1?
8) There is no quantification of the comparisons, i.e. no correlation coefficients, curve fits or other statistical tests applied to justify comments like "fine" or "reasonable. In fact, the authors' conclusions that the backscattering comparison is "fine", does not agree with what we see in the figures where the dispersion is hidden by the logarithmic scales on the figures.
9) I recommend that the analysis of the OPC data to derive backscattering should use the actual scattering measured by the OPC, rather than converting scattering to

equivalent optical diameters and then computing scattering. This adds additional uncertainty because there are large errors in size derivation because of Mie oscillations and unknown shape. If the authors derived backscatter from the measured forward scattering, as was done by Baumgardner and Clark (1998), this removes much of the inherent error.

10) I was disappointed by the excessive typographical and grammatical errors since the second author is a native English speaker. *"Author contributions. FC was responsible for most of the writing, **review and editing process, supported by all co-authors."** This appears to be inaccurate.

Reference

Baumgardner, D. and A. Clarke, 1998: Changes in aerosol properties with relative humidity in the remote southern hemisphere marine boundary layer, J. Geophys. Res., 103,16525-16534.

---

## Referee Comment (RC2)

**Review of Cairo et al., 2022**

The goal of this paper is to use in-situ and remote sensing measurements to test computations of optical properties of polar stratospheric clouds (PSCs). These computations are complicated since PSCs are composed not only of different sizes of particles, but also external mixtures of particles of different composition, phase and shape. This manuscript is based on a prior one that was reviewed but not accepted. One of those reviewers suggested a different modeling approach—employing a T-Matrix approach applicable to non-spherical particles. The authors have used that technique for the larger PSC particles (more likely to be aspherical) in this revised paper. While the results are not ideal, with some additional exploration and revision as discussed below, I believe the paper could make a useful contribution to the scientific community.

**Major Comments:**

A comparison of calculated to measured backscatter and depolarization yields disappointing results for depolarization. While some possible explanations are put forth, they are not well explored or connected to the characteristics of the PSCs sampled. The paper is relatively short, so there is plenty of room for more exploration and discussion. For example, the depolarization plots (Fig 5) do sometime show good agreement at low backscatter and high depolarization—it would be useful to examine these particular cases and see what is different. Under these conditions, what is really happening physically? Are these primarily large or small particles and what is the predominant phase/composition/ temperature that is contributing to agreement with the model? Likewise for the cases with poor agreement. For those cases, could a change in phase/shape/AR for some particles bring things into better accord? Plots of backscatter and depolarization for varying cases with associated size distributions (see below) could be helpful in determining what is causing the variations in model success. This would be useful for guiding future work by this group and others.

I also suggest that the authors show some particle size distributions to give an idea of the actual distribution of particles within the sampled PSCs. How well are they represented with a unimodal or bimodal lognormal distribution? Could differences in the fit contribute to some of the errors in calculated results?

What is the uncertainty in the size distributions themselves (as well as in the other measurements)? These are likely to have significant error as well, particularly for larger, aspherical particles. By including the measurement uncertainties and combining them with the model uncertainties such as in refractive index, you could add error bars to Figures 4 and 5 and have a better idea how the model is really performing.

Figs 4 and 5: Calculated vs measured backscatter coefficients are shown but there are no regressions to evaluate the goodness of fit. The plots at top and bottom of Fig 4 look identical except at low backscatter—is this correct? Also, the caption for Fig 5 says backscatter coefficient, when what is plotted is depolarization.

In addition, there were many spelling and grammar errors noted that should be corrected before publication. Some but not all are listed below.

**Minor Comments:**

Line 40/41 "viable solutions" should be "viable solution".

Line 63: "attempts with simple empirical models to use it to mimic depolarization from aspherical particles does a very poor job" should be "do a very poor job".

Line 104: "throughful revision" should be "thorough revision".

Line 124: "a total of 141 data point from the Antarctic and 332 data point from the Arctic flights". "Points" instead of "point".

Line 143: "fraction" is misspelled.

Line 163: "single scattering properties by randomly oriented spheroids" should be "single scattering properties *of* randomly oriented spheroids".

Line 190-192: "The backscattering itself reproduces the Mie results for size parameters below unity, then is progressively reduced to values that can even be one third of the Mie value when the particle size parameters is above ten, again this reduction depends in no simple way on the value of AR."

"parameters is" should be "parameter is".

"Again this reduction depends in no simple way on the value of AR" should be a separate sentence.

Line 220: "the analysis do suggest to maintain" should be "the analysis does suggest to maintain".

Line 221: "contrains" should be "constraints".

Line 262: "asimptotic value" should be this "asymptotic value".

Line 282: "all solid partiles" should be "all solid particles".

Line 285: "match" is misspelled.

Line 286/7: "Albeit it cannot be excluded a defective accuracy in depolarization data that prevent to fully demonstrate the validity of our model" is very awkward and should be reworded. In addition, some evidence that this is a reasonable statement should be included—were there problems with the depolarization data? What is its uncertainty?

Figure 1, 4 and  5 captions: "We reports data points" should be "We report data points".

Figure 6 caption: "historam" should be "histogram".

Figure captions: Sometimes "color" is used and sometimes "colour". In the text, "colour" is used.

---

## Author Comment (AC1)

**Response to Darrel Baumgardner.**

We thank the reviewer for his thorough examination of our manuscript. The reviewer's suggestions have led to a significant expansion of our work.

Below are the responses to the reviewer's comments, these latter partially reported (in bold) and indented for ease of reading. We have also reported, in italics and indented, the relevant additions/modifications to the manuscript.

**A)...I think that how they applied the code was also too simplistic, e.g. assuming that all the particles had the same AR rather than trying combinations of size-dependent AR.**

We have used a simplified assumption, given our limited knowledge of NAT crystallization. There has been speculation of anisotropic growth, favoring large asphericities when particles grow to large sizes, due to less depleted vapor mixing ratios close to the extremity of the crystals (Grothe et al., 2006). Also, various particle shapes and habits might coexist due to different nucleation and growth histories. Thus, it is certainly possible that a choice of particle size-dependent AR exists, which may improve the agreement with backscattering/depolarization measurements. However, such an agreement, if found, would be easily open to further criticism since there is no basis to make complicated assumptions about the size dependent AR. By suitably adjusting the AR, we would regard such a result as a selection of the most desirable outcome. For these reason, in looking for the AR intervals that best match the backscattering measurements, we prefer to consider only an average AR, as the most conservative assumption. Nevertheless, we have commented on such option in the paragraph 4 Discussion:

In the T-matrix theory, for fixed AR, the depolarization depends on the particle size and maximizes for particular sizes. There is certainly a way to assume a particle size-dependent AR in our PSDs so as to reconcile the computations with the observed values. However, such an approach would have little physical basis and could only be justified to maximize the agreement of calculations. Therefore, we have not explored this possibility further, although it is possible that our simplified hypothesis of a common AR for every particle may be the cause of the bad agreement between data and calculations in some case.

In the extension of our work to meet the reviewer's suggestion, we have explored the possibility of deriving the AR that best matches computation and experiment, on a case by case basis. This is the main upgrade of our work, stimulated by the revision process. In the new version, we considered that the previously presented  $\beta$  or  $\delta$  analysis with respect to (Rth, AR) can only constrain a range of Rth. In fact, from figures 2 and 3, you can see that Rth must be about 0.5-0.8  $\mu$ m, while the compatible ARs are all those between 0.3-0.5 and 1.5-3. Hence, at that stage, we acknowledge that a particular choice of AR is somewhat arbitrary.

We decided then to perform an additional simulation. We kept  $R_{th}$  fixed at values in the range 0.5-0.8  $\mu m$  and, for each of the experimental delta we looked for the AR that best matches the T-matrix calculations, in the ranges  $\,$  0.3-0.55 and 1.5-3. Then we selected the AR which gave the best match as  $R_{th}$  varied within 0.5-0.8  $\mu m$ .

Once found these AR, the same were used to compute  $\beta$  and compare it with its measured values, on a case-by-case basis.

This procedure has led to a net improvement in the agreement between experimental and calculated  $\delta s$ , while it has not changed appreciably the agreement between the  $\beta s$ .

We have described this new procedure in the Abstract:

The parameters  $R_{th}$  and AR of our model have been varied between 0.1 and 2  $\mu$ m and between 0.3 and 3, respectively, and the calculated backscattering coefficient and depolarization were compared with the observed ones.

The best agreement was found for  $R_{th}$  between 0.5 and 0.8  $\mu$ m, and for AR less than 0.55 and greater than 1.25.

To further constrain the variability of AR within the identified intervals we have sought an agreement with the experimental data by varying AR on a case-by-case basis, and further optimizing the agreement by a proper choice of AR smaller than 0.55 and greater than 1.5, and  $R_{th}$  within the interval 0.5 and 0.8  $\mu$ m. The ARs identified in this way cluster around the values 0.5 and 2.5.

In paragraph 2.3 Variability with the threshold radius Rth and Aspect Ratio AR:

The result of this study allows to identify only the best  $R_{th}$ , resulting around 0.5-0.8  $\mu$ m, while the ARs compatible with the measurements are all those between 0.3-0.55 and 1.5-3. To further constrain AR we have kept  $R_{th}$  at a fixed value, chosen between 0.5 and 0.8  $\mu$ m and changed this value with a 0.1  $\mu$ m step. For each of these fixed  $R_{th}$ , and separately for each PSD, we identified in the intervals (0.3-0.55), (1.5,3) the value of AR which best matched the observed  $\delta$  with its computed value. Finally, for each PSD we selected the pair  $R_{th}$  and AR which provided the best match. Once the ARs and  $R_{th}$  have been selected by forcing the agreement between the  $\delta_A$ , the same ones have been used for the calculation of the  $\beta_A$ .

Figures 4 and 5 have been upgraded with the result of this new approach:

Figure 4. Scatterplot of computed vs measured particle backscattering coefficients  $\beta_{A}$ . The ARs used for the computations have been selected, case by case, to produce the best

agreement between the  $\beta$  computed and measured, and are here represented in color coding. Only ARs in the intervals between 0.3 and 0.55, and between 1.5 and 3, have been considered.  $R_{th}$  was also selected within the interval 0.5-0.8  $\mu$ m to provide the best match. We report data points with BR greater than 1.2,  $\beta_{cross A}$  greater than 5 1026km21sr21 and temperature at the observation below 200 K.

Figure 5. Scatterplot of computed vs measured particle depolarization  $\delta_A$ . The ARs used for the computations are those that provided the best match between the  $\delta_A$  computed and measured, and are here represented in color coding. Only ARs in the intervals between 0.3 and 0.55, and between 1.5 and 3, have been considered. Rth was also selected within the interval 0.5-0.8  $\mu$ m to provide the best match.. We report data points with BR greater than 1.2,  $\beta_{cross}$  A greater than 5 10-6km-1sr-1 and temperature at the observation below 200 K.

We have also discussed the physical relevance of our results, in the par. 5. Discussion, which has been extensively rewritten. See our answer to "2)Diagnose their model results to understand why they are producing unacceptable comparisons."

**B) ... the authors never explain how their study will benefit the science, not in the introduction nor in the concluding remarks; hence, the questions are 1) What purpose does constraining the model parameters serve?**

Good point. We have not sufficiently placed our study in the broader perspective of PSC research. We have broadened the Introduction to include there the fact that the subsidence of large NAT particles is considered one of the main causes of denitrification of the polar winter stratosphere. Their settling time influences this process, which is in turn dependent on NAT particle shape and size, both of which determine their settling speed and lifetime, hence their denitrification efficacy. Of particular interest is the shape, given that non-spherical particles may fall significantly slower than volume equivalent spheres. A positive result of our study, which maybe we did not adequately underline, is that it strongly suggests avoiding ARs too close to 1.0, preferring ARs below 0.55 or above 1.5. This result has been underlined and referred to previous estimates on the asphericity of NAT particles.

We have reported such considerations in the Introduction:

The aims of this effort are both to verify the ability of the T-matrix approach to reproduce the observations from lidar/backscattersonde, once the PSDs are supposed known, and to provide a contribution to the estimation of the shape and size limits of the NAT PSC particles. The question of the shape of NAT particles is in fact far from being clarified, and has important implications for the denitrification mechanisms of the polar stratosphere, an important step in the process that lead to the destruction of stratospheric ozone. In fact, large PSC NAT particles settling down are considered one of the main causes of denitrification of the polar winter stratosphere (Di Liberto et al., 2015). Their settling time influences this process, and it is in turn dependent on NAT particles shape and size, both of which determining their settling speed and lifetime, hence their denitrification efficacy. Woiwode et al. (2014) assumed significantly non-spherical NAT particles to simulate the NAT settling speed leading to a the vertical redistribution of HNO3 observed between two companion flights during the RECONCILE airborne field campaign in the Arctic (von Hobe et al., 2013). Woiwode et al. (2016, 2019) have also suggested that NAT particles may be highly aspherical based on the infrared spectrometer MIPAS-STR limb observations exhibiting a spectral signature around 820 cm-1 and an overall spectral pattern compatible with large highly aspherical NAT particles. T-Matrix calculations assuming randomly oriented highly aspherical NAT particles (aspect ratios 0.1 or 10 for elongated or disk-like spheroids, respectively) were able to reproduce the MIPAS-STR observations to a large degree. Molleker et al. (2014) hypothesized strongly aspherical NAT particles to reconcile the amount of the condensed HNO3 resulting from PSC cloud spectrometer measurements with the expected stratospheric values, and to provide consistency between particles settling velocities and growth times with back trajectories. Moreover, Grothe et al. (2006) observed highly aspherical NAT in laboratory experiments. This is in contrast with earlier studies that assumed an AR = 0.9 for the NAT spheroids to match microphysical model simulation with airborne (Carslaw et al., 1998) or satellite borne (Hoyle et al., 2013; Engel et al., 2013) lidar observations.

**C) ...2)On what basis do they declare that these ranges are reasonable?**

As quoted above concerning NAT asphericities, Molleker et al. (2014) and Woiwode et al. (2014; 2016; 2019) have suggested that NAT particles may be highly aspherical.

Similarly, our study concludes that the best agreement between measurements and optical modeling occurs for strongly aspherical NAT.

This is in contrast with earlier studies that assumed an AR = 0.9 for NAT spheroids (Carslaw et al., 1998, Hoyle et al., 2013, Engel et al. 2013).

Concerning the  $R_{Th}$ , the hypothesis of dividing the PSD into a liquid part and a solid part on the basis of size is a hypothesis supported by what we know about PSC particle formation and

measurements (Deshler et al., 2003), leads to sensible results, and is in agreement with the depolarization-large particle correlation qualitatively presented in fig.1

**D*) ...3) Since they conclude that using prolate and oblate spheroids to model the scattering did not lead to useful results, that constraining the AR range is meaningless.**

Evidently we failed to convey our conclusion correctly. The study leads to three useful results: i. in an externally mixed PSC, it is reasonable to place a threshold radius  $R_{Th}$  around 0.6 µm, which divides the liquid part from the solid part of the particulate; ii. It is sensible to expect strongly aspherical shapes for the solid part of the particulate; iii. the observed depolarization is difficult to reproduce by a T-matrix approach. The latter can be considered a negative result, but it is a result nonetheless.

These considerations are reported in the Conclusion.

... our analysis has provided the range of optimal  $R_{th}$  and AR parameters that best match the observations. To sum up: i. in an externally mixed PSC, it is reasonable to place a threshold radius  $R_{th}$  between 0.5 and 0.8  $\mu$ m which divides the liquid part from the solid part of the particulate; ii. It is sensible to expect strongly aspherical shapes for the solid part of the cloud; iii. There are cases, in particular those related to high depolarization observations, in which, within our assumptions (i.e. a single form for the solid particulate, a fixed threshold radius for all PSD) prevents to reproduce the observed depolarization with a T-matrix approach.

In his suggestion to resubmit the manuscript, the reviewer asked to:

**1) Explain the importance of knowing the particle sizes and shapes in mixed phase PSCs.**

As outlined above, (see point A)) we have expanded the Introduction by placing our study in the broader perspective of PSC research.

**2) Diagnose their model results to understand why they are producing unacceptable comparisons.**

The novel approach we have pursued in the revision of our manuscript has led to new results. This has led us to a major revision of paragraph 4. Discussion. We report here the full text:

[revised manuscript text omitted]

---

## Author Comment (AC2)

**Response to Cynthia Twohy.**

We thank the reviewer for his thorough examination of our manuscript. The reviewer's suggestions have led to a significant revision and expansion of our work.

In the revision of our manuscript we have explored the possibility of deriving the AF that best matches computation and experiment, on a case by case basis. This is the main upgrade of our work. In this new version, we have acknowledged that the presented $\beta$ or $\delta$ measured-computed comparison with respect to ($R_{th}$, AR) can only constrain a range of $R_{th}$. In fact, from figures 2 and 3, you can see that $R_{th}$ must be about 0.5-0.8 $\mu$m, while the compatible ARs are all those between 0.3-0.5 and 1.5-3. Hence, at that stage, a particular choice of AR is somewhat arbitrary.
In the revision of the paper, we have performed additional simulations, and for each of the experimental (PSD, $\beta$, $\delta$) datum we have sought the AR and $R_{th}$ that best matches the measured $\delta$ with the T-matrix calculations. AR and $R_{th}$ were allowed to vary respectively in the ranges (0.3-0.6)U(1.5-3) and 0.5-0.8 $\mu$m.
Once these AR and $R_{th}$ have been found, the same were used to calculate $\beta$ and compare it with the experimental, on a case-by-case basis.
Of course, this procedure leads to an improvement in the agreement between experimental and calculated $\delta$s, while it does not change appreciably the agreement between the $\beta$s.

We have described this new procedure in the Abstract:

*The parameters $R_{th}$ and AR of our model have been varied between 0.1 and 2 $\mu$m and between 0.3 and 3, respectively, and the calculated backscattering coefficient and depolarization were compared with the observed ones.*
*The best agreement was found for $R_{th}$ between 0.5 and 0.8 $\mu$m, and for AR less than 0.55 and greater than 1.5.*
*To further constrain the variability of AR within the identified intervals we have sought an agreement with the experimental data by varying AR on a case-by-case basis, and further optimizing the agreement by a proper choice of AR smaller than 0.55 and greater than 1.5, and $R_{th}$ within the interval 0.5 and 0.8 $\mu$m. The ARs identified in this way cluster around the values 0.5 and 2.5.*

In paragraph 2.3 Variability with the threshold radius $R_{th}$ and Aspect Ratio AR:

*The result of this study allows to identify only the best $R_{th}$, resulting around 0.5-0.8 $\mu$m, while the ARs compatible with the measurements are all those between 0.3-0.55 and 1.5-3.*
*To further constrain AR we have kept $R_{th}$ at a fixed value, chosen between 0.5 and 0.8 $\mu$m and changed this value with a 0.1 $\mu$m step. For each of these fixed $R_{th}$, and separately for each PSD, we identified in the intervals (0.3-0.55), (1.5,3) the value of AR which best matched the observed $\delta$ with its computed value. Finally, for each PSD we selected the pair $R_{th}$ and AR which provided the best match. Once the ARs and $R_{th}$ have been selected by forcing the agreement between the $\delta_A$, the same ones have been used for the calculation of the $\beta_A$.*

Figures 4 and 5 have been upgraded with the result of this new approach:

[revised manuscript text omitted]

*Black dots represents the ARs providing the best match between the d and those computed from concomitant measurements of PSD.*

**the depolarization plots (Fig 5) do sometime show good agreement at low backscatter and high depolarization—it would be useful to examine these particular cases and see what is different. Under these conditions, what is really happening physically? Are these primarily large or small particles and what is the predominant phase/composition/ temperature that is contributing to agreement with the model? Likewise for the cases with poor agreement. For those cases, could a change in phase/shape/AR for some particles bring things into better accord? Plots of backscatter and depolarization for varying cases with associated size distributions (see below) could be helpful in determining what is causing the variations in model success.**

The 4. Discussion paragraph has been extensively rewritten and is now addressing more closely the reviewers remarks.

*In general, our model leads to good correlations between measured and modeled $\beta$s. For the $\delta$s the measurements are well reproduced by the calculations in many instances, as is the case for many of the selected ARs in the range 0.3-0.55. However, there are other cases in which the agreement is worse (when the best ARs have been selected in the range 1.5-3), or does not occur at all, as in the cases of observed depolarizations greater than 30%. In these latter cases, the impossibility of reproducing the observed values even under the hypothesis of a completely solid particles implies that, for those PSDs, our model is not able to produce the observed depolarizations. In these particular cases in which the model performs particularly badly, there may be problems of inhomogeneities of the cloud. These cases come from Antarctic observations, for which the microphysical observations from the balloon and the optical ones from ground-based lidar are separated geometrically, so that the two instruments sample air masses separated by several tens of kilometres, and it may be the case that some clouds were not homogeneous on such spatial scales.*

For what concerns the possibility of implementing a size-dependent AR approach, we write:

*Different shapes produce different polarization, according to T-Matrix. This has also been proven experimentally since the early work of Sassen and Hsueh (1998) and Freudenthaler et al. (1996) that showed how lidar depolarization ratios in persisting contrails ranged from 10% to 70%, depending on the stage of their growth and on temperature. In the T-matrix theory, for fixed AR, the depolarization depends on the particle size and maximizes for particular sizes. There is certainly a way to assume a particle size-dependent AR in our PSDs so as to reconcile the computations with the observed values. However, such an approach would have little physical basis and could only be justified to maximize the agreement of calculations. Therefore, we have not explored this possibility further, although it is possible that our simplified hypothesis of a common AR for every particle may be the cause of the bad agreement between data and calculations in some case.*

**I also suggest that the authors show some particle size distributions to give an idea of the actual distribution of particles within the sampled PSCs.**

Concerning the shape of the PSDs which is suggested to display, our study is based on 473 data points (i.e. 473 triplets of PSD, backscattering coefficient, and depolarization). A three panel plot showing: i. the time series of PSD (color plot), ii. the corresponding backscatter coefficient (line plot); iii. the corresponding depolarization (line plot) could be produced, but given the range of the observations such representation may be difficult to interpret. However, in the revision of the manuscript, when discussing the AR distribution in new Figure 6, we have correlated it with a parameter characterizing the shape of the PSD.

**How well are they represented with a unimodal or bimodal lognormal distribution? Could differences in the fit contribute to some of the errors in calculated results? What is the uncertainty in the size distributions themselves (as well as in the other measurements)? These are likely to have significant error as well, particularly for larger, aspherical particles. By including the measurement uncertainties and combining them with the model uncertainties such as in refractive index, you could add error bars to Figures 4 and 5 and have a better idea how the model is really performing.**

The treatment of uncertainty has been elaborated, and error bars have been used in the new figures 4 and 5. Errors have been discussed in 2.1 Dataset:

> *Experimental errors in the particle Backscatter Ratio (R-1) are estimated to be 5%, but not less than 0.05 in absolute value, while the error in volume depolarization is about 10%–15%. Additional uncertainty comes from the determination of pressure and temperature from radiosoundings, needed to compute $\beta_A$ and $\delta_A$ (Adriani et al., 2004).*
> *[...]*
> *Particle size histograms are fitted to unimodal or bimodal lognormal size distributions, which are the representation of size distribution used in this work. The uncertainties on the determination of the parameters of the mono/bimodal lognormals were determined by Deshler et al. (2003b) with Monte Carlo simulations. These were 20% for distribution width, 30% for median radii and 10% for modal particle concentrations.*

And in 3. Results:

*The uncertainties associated with the measured $\beta_A$ and $\delta_A$ derive from the error analysis for the single lidar data, which can be found in Adriani et al. (2004) or from the standard deviation for the averaged data, depending on which is greater. The uncertainties on the calculated $\beta_A$ and $\delta_A$, are 40% as determined by Deshler et al. (2003a) for any moment of a PSD derived from the OPC measurements. Deshler et al. determined this through a Monte Carlo simulation which used the uncertainties of the OPC size and concentration measurements to quantify the uncertainties in the PSD parameters and their subsequent moments.*

**Figs 4 and 5: Calculated vs measured backscatter coefficients are shown but there are no regressions to evaluate the goodness of fit.**

We have now provided the Pearson correlation coefficient (resulting to be 0.56) for the goodness of the 1:1 fit for the β comparison and added 1:1 lines to the data in Figs. 4 and 5.

We did not perform goodness-of-fit tests for the comparison of δs. In this case it is clear that there is a set of well-aligned points along the 1:1 line, and sets of points that deviate from it in a nonrandom way. We have discussed the different characteristics of these sets in the 3 Result paragraph, totally rewritten, which we report here in its entirety:

**3 Results**

*Figure 4 reports the scatterplot of measured vs computed $\beta_A$, colour coded in terms of AR. The figure represents the analogue of Figure 4 in Snels et al. (2021), where in the present case we have used a larger dataset, including now four Arctic balloon flights, and used T-Matrix instead of a factor 0.5 reduction in the Mie backscattering. Figure 5 reports the scatterplot of measured vs computed $\delta_A$ similarly color coded in terms of AR. The uncertainties associated with the measured $\beta_A$ and $\delta_A$ derive from the error analysis for the single lidar data, which can be found in Adriani et al. (2004) or from the standard deviation for the averaged data, depending on which is greater. The uncertainties associated with the measured $\beta_A$ and $\delta_A$ derive from the error analysis for the single lidar data, which can be found in Adriani et al. (2004) or from the standard deviation for the averaged data, depending on which is greater. The uncertainties on the calculated $\beta_A$ and $\delta_A$, are 40% as determined by Deshler et al. (2003a) for any moment of a PSD derived from the OPC measurements. Deshler et al. determined this through a Monte Carlo simulation which used the uncertainties of the OPC size and concentration measurements to quantify the uncertainties in the PSD parameters and their subsequent moments.*

*Despite the dispersion in Figure 4 the points cluster around the straight line $\beta_{calc}=\beta_{meas}$, indicating the agreement between computation and measurements can be considered fine for $\beta_A$ with the exception of $\beta$ values below $4 \ 10^{-5} km^{-1} sr^{-1}$ where the $\beta_{calc}$ underestimate the measurements. Such underestimation seems to be of the order of $10^{-5} km^{-1} sr^{-1}$, a magnitude compatible with possible inaccuracies in the calibration of the lidar data. The Pearson correlation coefficient for the entire dataset is 0.56, and increases if the lower values of $\beta$ are neglected.*

*The $\delta_A$ scatterplot shows the presence of a good number of points that align along the $\delta_{calc}=\delta_{meas}$ correlation line, with AR selected mainly around the value 0.5. However, for depolarization values greater than 30% there is no AR that will reproduce the measurements. These points correspond to those presented in Figure 1, with low values of BR and high values of the concentration ratio of large to total particles. They mainly come from three single observational periods of about one minute each, characterized by air temperatures between 184-188 K. Given the magnitude of the depolarization, it is possible that those observations are not referable to clouds in mixed phase, but rather to clouds of predominantly solid particles. For that particular set of points, we also explored the possibility that all particles were solid, but even under this assumption the comparison with the experimental data did not improve appreciably.*

*In Figure 5 for depolarizations lower than 15%, the points which deviate, by excess or defect, from the 1:1 straight line have predominantly AR greater than 1.5. So it seems that selected ARs greater than 1.5 generally produce a worse correlation. From Figure 4 we observe that AR values in the range (0.3-0.55) tend to be associated with medium-low $\beta$ values, while AR values in the range (1.5-3) are mainly associated with medium-high $\beta$.*

*To conclude, the choice of $R_{th}$ in a range between 0.5 and 0.8 $\mu m$ leads to a reasonably good agreement between the $\beta$'s, but there seems to be a discrepancy between the calculated value and the measurements in their lower range of variability.*

*From Figure 4 such mismatch, which makes the measurements larger than the calculations, seems to be of the order of $10^{-5} km^{-1} sr^{-1}$. The selection of the AR that produces the best agreement with the observed $\delta$'s leads to three results: i. The ARs in the range 0.3-0.55 tend to be selected in correspondence with medium-low $\beta$'s, the ARs in the range 1.5-3 in*

*correspondence with medium-high $\beta$'s. ii. ARs in the 0.3-0-5 range reproduce the measurements well, except for some observations where the depolarizations are greater than 30%; iii. the ARs in the 1.5-3 range reproduce the measurements less well; iv. There is no AR that will allow the calculations to reproduce the measurements for depolarizations greater than 30%.*

**The plots at top and bottom of Fig 4 look identical except at low backscatter—is this correct?**

Yes, it was. However, we have updated figure 4 with the new results.

**the caption for Fig 5 says backscatter coefficient, when what is plotted is depolarization.**

We are sorry for this. We have corrected our mistake.

Finally, we wish to thank the reviewer for her patience in spotting out all our typographical and grammatical errors. We apologize for the poor quality of the written English, responsibility of the first author only. This has been corrected in the revised manuscript.

---

## Author Comment (AC3)

**Response to Darrel Baumgardner.**

We thank the reviewer for his thorough examination of our manuscript. The reviewer's suggestions have led to a significant expansion of our work.

Below are the responses to the reviewer's comments, these latter partially reported (in bold) and indented for ease of reading. We have also reported, in italics and indented, the relevant additions/modifications to the manuscript.

> **A)...I think that how they applied the code was also too simplistic, e.g. assuming that all the particles had the same AR rather than trying combinations of size-dependent AR.**

We have used a simplified assumption, given our limited knowledge of NAT crystallization. There has been speculation of anisotropic growth, favoring large asphericities when particles grow to large sizes, due to less depleted vapor mixing ratios close to the extremity of the crystals (Grothe et al., 2006). Also, various particle shapes and habits might coexist due to different nucleation and growth histories. Thus, it is certainly possible that a choice of particle size-dependent AR exists, which may improve the agreement with backscattering/depolarization measurements. However, such an agreement, if found, would be easily open to further criticism since there is no basis to make complicated assumptions about the size dependent AR. By suitably adjusting the AR, we would regard such a result as a selection of the most desirable outcome.
For these reason, in looking for the AR intervals that best match the backscattering measurements, we prefer to consider only an average AR, as the most conservative assumption. Nevertheless, we have commented on such option in the paragraph 4 Discussion:

> *In the T-matrix theory, for fixed AR, the depolarization depends on the particle size and maximizes for particular sizes. There is certainly a way to assume a particle size-dependent AR in our PSDs so as to reconcile the computations with the observed values. However, such an approach would have little physical basis and could only be justified to maximize the agreement of calculations. Therefore, we have not explored this possibility further, although it is possible that our simplified hypothesis of a common AR for every particle may be the cause of the bad agreement between data and calculations in some case.*

In the extension of our work to meet the reviewer's suggestion, we have explored the possibility of deriving the AR that best matches computation and experiment, on a case by case basis. This is the main upgrade of our work, stimulated by the revision process. In the new version, we considered that the previously presented $\beta$ or $\delta$ analysis with respect to ($R_{th}$, AR) can only constrain a range of $R_{th}$. In fact, from figures 2 and 3, you can see that $R_{th}$ must be about 0.5-0.8 $\mu$m, while the compatible ARs are all those between 0.3-0.5 and 1.5-3. Hence, at that stage, we acknowledge that a particular choice of AR is somewhat arbitrary.
We decided then to perform an additional simulation. We kept $R_{th}$ fixed at values in the range 0.5-0.8 $\mu$m and, for each of the experimental delta we looked for the AR that best matches the T-matrix calculations, in the ranges 0.3-0.55 and 1.5-3. Then we selected the AR which gave the best match as $R_{th}$ varied within 0.5-0.8 $\mu$m .
Once found these AR, the same were used to compute $\beta$ and compare it with its measured values, on a case-by-case basis.

This procedure has led to a net improvement in the agreement between experimental and calculated δs, while it has not changed appreciably the agreement between the βs.

We have described this new procedure in the Abstract:

*The parameters $R_{th}$ and AR of our model have been varied between 0.1 and 2 μm and between 0.3 and 3, respectively, and the calculated backscattering coefficient and depolarization were compared with the observed ones.*

*The best agreement was found for $R_{th}$ between 0.5 and 0.8 μm, and for AR less than 0.55 and greater than 1.25.*

*To further constrain the variability of AR within the identified intervals we have sought an agreement with the experimental data by varying AR on a case-by-case basis, and further optimizing the agreement by a proper choice of AR smaller than 0.55 and greater than 1.5, and $R_{th}$ within the interval 0.5 and 0.8 μm. The ARs identified in this way cluster around the values 0.5 and 2.5.*

In paragraph 2.3 Variability with the threshold radius $R_{th}$ and Aspect Ratio AR:

*The result of this study allows to identify only the best $R_{th}$, resulting around 0.5-0.8 μm, while the ARs compatible with the measurements are all those between 0.3-0.55 and 1.5-3.*

*To further constrain AR we have kept $R_{th}$ at a fixed value, chosen between 0.5 and 0.8 μm and changed this value with a 0.1 μm step. For each of these fixed $R_{th}$, and separately for each PSD, we identified in the intervals (0.3-0.55), (1.5,3) the value of AR which best matched the observed δ with its computed value. Finally, for each PSD we selected the pair $R_{th}$ and AR which provided the best match. Once the ARs and $R_{th}$ have been selected by forcing the agreement between the $δ_A$, the same ones have been used for the calculation of the $β_A$.*

Figures 4 and 5 have been upgraded with the result of this new approach:

[Figure]

*Figure 4. Scatterplot of computed vs measured particle backscattering coefficients $β_A$. The ARs used for the computations have been selected, case by case, to produce the best*

*agreement between the β computed and measured, and are here represented in color coding. Only ARs in the intervals between 0.3 and 0.55, and between 1.5 and 3, have been considered. $R_{th}$ was also selected within the interval 0.5-0.8 μm to provide the best match. We report data points with BR greater than 1.2, $β_{cross\ A}$ greater than 5 10⁻6km⁻1sr⁻1 and temperature at the observation below 200 K.*

[Figure]

*Figure 5. Scatterplot of computed vs measured particle depolarization $δ_A$. The ARs used for the computations are those that provided the best match between the $δ_A$ computed and measured, and are here represented in color coding. Only ARs in the intervals between 0.3 and 0.55, and between 1.5 and 3, have been considered. $R_{th}$ was also selected within the interval 0.5-0.8 μm to provide the best match.. We report data points with BR greater than 1.2, $β_{cross}$ A greater than 5 10⁻⁶km⁻¹sr⁻¹ and temperature at the observation below 200 K.*

We have also discussed the physical relevance of our results, in the par. 5. Discussion, which has been extensively rewritten. See our answer to "2)Diagnose their model results to understand why they are producing unacceptable comparisons."

**B) … the authors never explain how their study will benefit the science, not in the introduction nor in the concluding remarks; hence, the questions are 1) What purpose does constraining the model parameters serve?**

Good point. We have not sufficiently placed our study in the broader perspective of PSC research. We have broadened the Introduction to include there the fact that the subsidence of large NAT particles is considered one of the main causes of denitrification of the polar winter stratosphere. Their settling time influences this process, which is in turn dependent on NAT particle shape and

size, both of which determine their settling speed and lifetime, hence their denitrification efficacy. Of particular interest is the shape, given that non-spherical particles may fall significantly slower than volume equivalent spheres. A positive result of our study, which maybe we did not adequately underline, is that it strongly suggests avoiding ARs too close to 1.0, preferring ARs below 0.55 or above 1.5. This result has been underlined and referred to previous estimates on the asphericity of NAT particles.

We have reported such considerations in the Introduction:

> *The aims of this effort are both to verify the ability of the T-matrix approach to reproduce the observations from lidar/backscattersonde, once the PSDs are supposed known, and to provide a contribution to the estimation of the shape and size limits of the NAT PSC particles. The question of the shape of NAT particles is in fact far from being clarified, and has important implications for the denitrification mechanisms of the polar stratosphere, an important step in the process that lead to the destruction of stratospheric ozone. In fact, large PSC NAT particles settling down are considered one of the main causes of denitrification of the polar winter stratosphere (Di Liberto et al., 2015). Their settling time influences this process, and it is in turn dependent on NAT particles shape and size, both of which determining their settling speed and lifetime, hence their denitrification efficacy. Woiwode et al. (2014) assumed significantly non-spherical NAT particles to simulate the NAT settling speed leading to a the vertical redistribution of $HNO_3$ observed between two companion flights during the RECONCILE airborne field campaign in the Arctic (von Hobe et al., 2013). Woiwode et al. (2016, 2019) have also suggested that NAT particles may be highly aspherical based on the infrared spectrometer MIPAS-STR limb observations exhibiting a spectral signature around 820 $cm^{-1}$ and an overall spectral pattern compatible with large highly aspherical NAT particles. T-Matrix calculations assuming randomly oriented highly aspherical NAT particles (aspect ratios 0.1 or 10 for elongated or disk-like spheroids, respectively) were able to reproduce the MIPAS-STR observations to a large degree. Molleker et al. (2014) hypothesized strongly aspherical NAT particles to reconcile the amount of the condensed HNO3 resulting from PSC cloud spectrometer measurements with the expected stratospheric values, and to provide consistency between particles settling velocities and growth times with back trajectories. Moreover, Grothe et al. (2006) observed highly aspherical NAT in laboratory experiments. This is in contrast with earlier studies that assumed an AR = 0.9 for the NAT spheroids to match microphysical model simulation with airborne (Carslaw et al., 1998) or satellite borne (Hoyle et al., 2013; Engel et al., 2013) lidar observations.*

**C) …2)On what basis do they declare that these ranges are reasonable?**

As quoted above concerning NAT asphericities, Molleker et al. (2014) and Woiwode et al. (2014; 2016; 2019) have suggested that NAT particles may be highly aspherical.

Similarly, our study concludes that the best agreement between measurements and optical modeling occurs for strongly aspherical NAT.

This is in contrast with earlier studies that assumed an AR = 0.9 for NAT spheroids (Carslaw et al., 1998, Hoyle et al., 2013, Engel et al. 2013).

Concerning the $R_{Th}$, the hypothesis of dividing the PSD into a liquid part and a solid part on the basis of size is a hypothesis supported by what we know about PSC particle formation and

measurements (Deshler et al., 2003), leads to sensible results, and is in agreement with the depolarization-large particle correlation qualitatively presented in fig.1

> **D) …3) Since they conclude that using prolate and oblate spheroids to model the scattering did not lead to useful results, that constraining the AR range is meaningless.**

Evidently we failed to convey our conclusion correctly. The study leads to three useful results: i. in an externally mixed PSC, it is reasonable to place a threshold radius $R_{Th}$ around 0.6 μm, which divides the liquid part from the solid part of the particulate; ii. It is sensible to expect strongly aspherical shapes for the solid part of the particulate; iii. the observed depolarization is difficult to reproduce by a T-matrix approach. The latter can be considered a negative result, but it is a result nonetheless.

These considerations are reported in the Conclusion.

> *… our analysis has provided the range of optimal $R_{th}$ and AR parameters that best match the observations. To sum up: i. in an externally mixed PSC, it is reasonable to place a threshold radius $R_{th}$ between 0.5 and 0.8 μm which divides the liquid part from the solid part of the particulate; ii. It is sensible to expect strongly aspherical shapes for the solid part of the cloud; iii. There are cases, in particular those related to high depolarization observations, in which, within our assumptions (i.e. a single form for the solid particulate, a fixed threshold radius for all PSD) prevents to reproduce the observed depolarization with a T-matrix approach.*

In his suggestion to resubmit the manuscript, the reviewer asked to:

> **1)      Explain the importance of knowing the particle sizes and shapes in mixed phase PSCs.**

As outlined above, (see point A)) we have expanded the Introduction by placing our study in the broader perspective of PSC research.

> *2)*      **Diagnose their model results to understand why they are producing unacceptable comparisons**.

The novel approach we have pursued in the revision of our manuscript has led to new results. This has led us to a major revision of paragraph 4. Discussion. We report here the full text:

[revised manuscript text omitted]

For completeness we report also new Figure 7.

[Figure]

*Figure 7: Sequences of $\beta$ (red dots) and $\delta$ (blue dots) measured on a balloon flight on December 9th 2001, from Kiruna, Sweden. Each data point represents an average over 60s. Black dots represents the ARs providing the best match between the $\delta$ and those computed from concomitant measurements of PSD.*

3) **Run a sensitivity analysis using simulated PSCs and measurements to quantify the observed difference.**

In a sense, the sensitivity of the method could have been estimated from the results already presented. Specifically, the sensitivity of our method can be obtained from the range of variability of the RMSEs according to the variability of AR and $R_{Th}$, as shown in figures 2 and 3. We note that

those results are relative to the analysis of real particle size distributions (PSDs), from measurements. We don't see how a similar study, done on simulated PSDs, would provide additional information.

However, to pursue the reviewer's requests in a different sense, we have estimated the uncertainties to be attributed to both computed and measured $\beta$s and $\delta$s (see new figures 4 and 5); We have added in the revised par. 3 Results:

> *The uncertainties associated with the measured $\beta_A$ and $\delta_A$ derive from the error analysis for the single lidar data, which can be found in Adriani et al. (2004) or from the standard deviation for the averaged data, depending on which is greater. The uncertainties on the calculated $\beta_A$ and $\delta_A$, are 40% as determined by Deshler et al. (2003a) for any moment of a PSD derived from the OPC measurements. Deshler et al. determined this through a Monte Carlo simulation which used the uncertainties of the OPC size and concentration measurements to quantify the uncertainties in the PSD parameters and their subsequent moments.*

**4) Correct a large number of typographical and grammatical errors that made the current manuscript distractive to read.**

We apologize for the poor quality of the written English, responsibility of the first author only. This has been corrected in the revised manuscript.

Other comments, questions and suggestions:

1) **From the abstract onward the authors erroneously talk about comparing "microphysical and optical" measurements. This makes no sense since all of the measurements are microphysical and optical, i.e. the OPC uses an optical technique to derive size distributions that help describe the microphysical properties of the PSCs. Likewise, the remote sensing techniques are optical and are also used to derive microphysical properties of PSCs.**

This seems to be splitting hairs. If we are not mistaken, almost every cloud probe available today uses an optical technique to measure cloud and aerosol size distributions. The days of impactors has long since faded into history. Yet the results of the aircraft optical probes are used to discuss cloud and aerosol microphysics. It is not clear to us that we are doing anything different, as long as the uncertainties inherent in the instrument, due to its use of optics to make the measurement, is clearly described, as it is here and in the referencing literature. Generally, when an instrument provides size distributions it is discussed as a microphysical measurement, not an optical measurement, even though the fundamental principle on which the measurements is made is optical. The lidar and backscattersonde provide less detailed microphysical measurements, but there is still microphysical information contained therein, such as the extent of aspherical particles which is a microphysical property measured optically. Airborne lidars are routinely used to measure cloud base, cloud top, and the extent of ice in the cloud, all microphysical as well as optical properties.

2) **The authors never explain the relative importance of mixture of particle types in mixed phase PSCs. Had the modeling exercise been successful, who would benefit?**

There are mainly three goals. To test the ability of the T-matrix code to reproduce the observations from lidar/backscattersonde, once the PSDs are known. To provide an estimate of the AR parameters and of the smallest dimensions of the solid part of the PSC particulate mixture. These goals are now reported more clearly in the Introduction.

3) **Nothing is discussed about the contribution to the backscattering of other types of stratospheric particles, e.g., meteoritic dust, sulfate particles, etc. How does that impact the measurements and modeling?**

The study exploits measurements taken within PSCs. The contribution to backscattering and depolarization of the background atmospheric particulate matter (SSA, meteoric dust, etc.), possibly observable in isolation outside the cloud, is negligible within a cloud in the majority of cases under exam. However, it is possible that unaccounted background aerosol led to unaccurate lidar calibration. We acknowledge this fact in the 4 Result paragraph.

*Despite the dispersion in Figure 4 the points cluster around the straight line $\beta_{calc}=\beta_{meas}$, indicating the agreement between computation and measurements can be considered fine for $\beta$ with the exception of $\beta$ values below 4   $10^{-5}km^{-1}sr^{-1}$ where $\beta_{calc}$ underestimate the measurements. Such underestimation seems to be of the order of $10^{-5}km^{-1}sr^{-1}$, of the same order of the backscattering from the background atmospheric particulate matter in volcanic quiescent conditions, a magnitude compatible with possible inaccuracies in the calibration of the lidar data.*

4) **The backscatter instrument described by Adriani (1999) had multiple wavelengths. Why is only the 532 being used? Wouldn't modeling multiple wavelengths have improved the retrievals?**

We compare PSD mainly with measurements from a lidar which does not have the second wavelength. The backscattersonde has also been used on a part of the dataset and it provided measurements at a second wavelength, but they have not been judged accurate enough to be published.

5) **If this was a true modeling study, an iterative methodology should have been used to vary the mixtures of shapes and sizes until most closely matched by the measurements.**

In our study both the AR and the $R_{Th}$ were independently varied in order to simulate different mixtures of shapes and sizes, thus the parameter ranges that best matches the measures were identified. Figures 2 and 3 in our manuscript provide what the reviewer is asking here. This analysis was effective in delimiting the variability of $R_{th}$, but not of AR, therefore in the revision of the manuscript we proceeded to look for the best match on a case-by-case basis, identifying for each PSD the best AR within the wide ranges of variability previously identified. The result is, for $R_{th}$ within the limits 0.5-0.8 $\mu$m,  a distribution of AR within these intervals, which clusters around the values 0.5 and 2.5, approximately, as reported in the new figure 6 where the AR distribution is reported in terms of a parameter characterizing the shape of the PSD.

[Figure]

Figure 6. 2D-histogram of occurrence of ARs and of N(r < 0.7μm)=N_tot, the ratio between particles with radius greater than 0.7 μm and total particles. Only ARs in the intervals between 0.3 and 0.55, and between 1.5 and 3, have been considered.

6) **How homogeneous are these clouds and what do the PSDs look like derived from the OPC? The reader never sees the actual shapes of PSD or what the number concentrations are. This is important because it will impact the backscattering and depolarization. It is stated in the results section that apparently the larger particles are biasing the depolarization but this depends on the total concentration of particles and how homogeneous the mixture is. I could not find in the Adriani (1999) paper what beam volume is at each measurement gate.**

Answering the last question first. The backscattersonde laser beam cross section is approximately 20 mm $^2$, and 90% of the backscattered signal comes for 2 to 6 meters from the backscattersonde, so the sampled volume is of the order of 50-100 cm$^3$.

The question of homogeneity of PSCs is relevant. Our database is composed of two groups of measurements. In the first, the OPC on board a balloon is compared with Antarctic ground-based lidar measurements. In the second, the OPC is compared with Arctic measurements from a backscattersonde on board the same balloon. It is clear that the second class of measurements is less affected by cloud inhomogeneity problems, given that both instruments measure in-situ the same cloud.

Opposed to this is the first group where measurements taken in cloud regions are by their nature separated in space and time as the balloon drifts down range from the lidar, so cloud inhomogeneity must be considered. This issue was addressed in Snels et al., (2021), a work based on the same Antarctic measurements. Snels et al. compared the lidar profiles with backscattering computations from the OPC data. Of the 18 coincident lidar-balloon flights, only 15 profiles were

used for the analysis. The choice was based on a visual inspection of the coincidence of the main cloud features in the lidar and balloon flight altitude profiles.

As outlined above, the present work adds to the Snels et al. (2021) Antarctic dataset, measurements from Arctic balloonborne OPC and backscattersonde flights. This addition does not alter significantly the goodness of the computed vs measured backscattering regression line. This gives us confidence that cloud inhomogeneity did not play a significant role in causing the dispersion of the points in the regression line.

However, for some data points for which the computed vs measured do not match at all, we have invoked a possible inhomogeneity of the cloud as a possible explaination. This has been reported in the last lines of the 4. Discussion paragraph (see answer to point 2) above).

Concerning the shape of the PSDs which is suggested to display, our study is based on 473 data points (i.e. 473 triplets of PSD, backscattering coefficient, and depolarization). Given the range of the observations, it is difficult to provide a representation of the actual shapes of PSD or what the number concentrations are. In any case we proceeded to use the ratio $N(r>6\mu m)/N_{tot}$ in the new figure 6 as a parameter to characterize the shape of the PSD. We have also checked a lack of clear correlation of $N_{tot}$ with the result of the model-measurement comparison.

7) **In Figs. 4 and 5, there is no noticeable difference between AR-0.5 and AR=1.5. This does not surprise me because if you have an ensemble of randomly oriented spheroid, an oblate spheroid will look like a prolate spheroid, depending on their relative orientations; hence why even use ARs < 1?**

It is certainly true that there are particular orientations for which prolate spheroids can appear like oblate spheroids, and vice versa. However, two PSDs with identical distribution parameters, one composed of oblate and the other of prolate spheroids, with reverse ARs, randomly oriented, need not necessarily have the same backscattering properties. That this is the case can be deduced, for instance, from Figure 1 in the work of Liu and Mishchenko (2001). However, old Figures 4 and 5 have been discarded.

In the new approach pursued following the reviewer' remarks, the difference in behavior between ARs less than or greater than 1 is more apparent.

8) **There is no quantification of the comparisons, i.e. no correlation coefficients, curve fits or other statistical tests applied to justify comments like "fine" or "reasonable. In fact, the authors' conclusions that the backscattering comparison is "fine", does not agree with what we see in the figures where the dispersion is hidden by the logarithmic scales on the figures.**

Good point. We have provided quantitative data on the goodness of the fit for the $\beta$ comparison (the Pearson correlation coefficient) and added 1:1 lines to the data in Figs. 4 and 5.

We did not perform goodness-of-fit tests for comparison of $\delta$s. In this case it is clear that there is a set of well-aligned points along the 1:1 line, and sets of points that significantly deviate from it in a non-random way. We have discussed the different characteristics of these sets in the 3 Result paragraph, totally rewritten, which we report here in its entirety:

*3 Results*

*Figure 4 reports the scatterplot of measured vs computed $\beta_A$, colour coded in terms of AR. The figure represents the analogue of figure 4 in Snels et al. (2021), where in the present case*

we have used a larger dataset, including now four Arctic balloon flights, and used T-Matrix instead of a factor 0.5 reduction in the Mie backscattering. Figure 5 reports the scatterplot of measured vs computed $\delta_A$ similarly color coded in terms of AR. The uncertainties associated with the measured $\beta_A$ and $\delta_A$ derive from the error analysis for the single lidar data, which can be found in Adriani et al. (2004) or from the standard deviation for the averaged data, depending on which is greater. The uncertainties associated with the measured $\beta_A$ and $\delta_A$ derive from the error analysis for the single lidar data, which can be found in Adriani et al. (2004) or from the standard deviation for the averaged data, depending on which is greater. The uncertainties on the calculated $\beta_A$ and $\delta_A$, are 40% as determined by Deshler et al. (2003a) for any moment of a PSD derived from the OPC measurements. Deshler et al. determined this through a Monte Carlo simulation which used the uncertainties of the OPC size and concentration measurements to quantify the uncertainties in the PSD parameters and their subsequent moments.

Despite the dispersion in Figure 4 the points cluster around the straight line $\beta_{calc}=\beta_{meas}$, indicating the agreement between computation and measurements can be considered fine for $\beta_A$ with the exception of $\beta$ values below 4 $10^{-5}km^{-1}sr^{-1}$ where the $\beta_{calc}$ underestimate the measurements. Such underestimation seems to be of the order of $10^{-5}km^{-1}sr^{-1}$, a magnitude compatible with possible inaccuracies in the calibration of the lidar data. The Pearson correlation coefficient for the entire dataset is 0.56, and increases if the lower values of $\beta$ are neglected.

The $\delta_A$ scatterplot shows the presence of a good number of points that align along the $\delta_{calc}=\delta_{meas}$ correlation line, with AR selected mainly around the value 0.5. However, for depolarization values greater than 30% there is no AR that will reproduce the measurements. These points correspond to those presented in Figure 1, with low values of BR and high values of the concentration ratio of large to total particles. They mainly come from three single observational periods of about one minute each, characterized by air temperatures between 184-188 K. Given the magnitude of the depolarization, it is possible that those observations are not referable to clouds in mixed phase, but rather to clouds of predominantly solid particles. For that particular set of points, we also explored the possibility that all particles were solid, but even under this assumption the comparison with the experimental data did not improve appreciably.

In Figure 5 for depolarizations lower than 15%, the points which deviate, by excess or defect, from the 1:1 straight line have predominantly AR greater than 1.5. So it seems that selected ARs greater than 1.5 generally produce a worse correlation. From Figure 4 we observe that AR values in the range (0.3-0.55) tend to be associated with medium-low $\beta$ values, while AR values in the range (1.5-3) are mainly associated with medium-high $\beta$.

To conclude, the choice of $R_{th}$ in a range between 0.5 and 0.8 $\mu m$ leads to a reasonably good agreement between the $\beta$'s, but there seems to be a discrepancy between the calculated value and the measurements in their lower range of variability.

From Figure 4 such mismatch, which makes the measurements larger than the calculations, seems to be of the order of $10^{-5}km^{-1}sr^{-1}$. The selection of the AR that produces the best agreement with the observed $\delta$'s leads to three results: i. The ARs in the range 0.3-0.55 tend to be selected in correspondence with medium-low $\beta$'s, the ARs in the range 1.5-3 in correspondence with medium-high $\beta$'s. ii. ARs in the 0.3-0-5 range reproduce the measurements well, except for some observations where the depolarizations are greater than 30%; iii. the ARs in the 1.5-3 range reproduce the measurements less well; iv. There is no AR that will allow the calculations to reproduce the measurements for depolarizations greater than 30%.

**9)** ***I recommend that the analysis of the OPC data to derive backscattering should use the actual scattering measured by the OPC, rather than converting scattering to equivalent optical diameters and then computing scattering. This adds additional uncertainty because there are large errors in size derivation because of Mie oscillations and unknown shape. If the authors derived backscatter from the measured forward scattering, as was done by Baumgardner and Clark (1998), this removes much of the inherent error.***

In Baumgardner and Clarke (1998) the authors infer the total single particle scattering coefficient from the forward scattering coefficient measured between 4 and 12 degrees. This inference is made by calculating, with the aid of Mie's theory, the relationship between the scattering, calculated in the above angle interval, and the total scattering. The inference is then that the total particle scattering can be inferred from the FSSP measured scattering, seemingly on a particle by particle basis. The OPC we employ is not a single particle scatterer, but rather discriminator levels are used to collect all photo multiplier pulses larger than a preset level. Thus the number concentration in any discriminator bracket is the result of all particles which provide a light signal above the lower level and less than the next discriminator level. These OPCs measure a maximum of 12 sizes between 0.19 and 10.0 µm, much less than the FSSP, and thus require fitting of size distributions to obtain estimates of backscatter. Estimating backscatter from 12 sizes would be insufficient to compare with the lidar measurements which are inherently ensemble measurements. In addition, the OPC uses white light to measure scattering at 40 degrees, an optical signal not directly comparable to the lidar. Furthermore, instead of using Mie theory, we should use T-matrix calculations and replicate them for many different AR and $R_{Th}$.

**10)** ***I was disappointed by the excessive typographical and grammatical errors since the second author is a native English speaker. "Author contributions. FC was responsible for most of the writing, review and editing process, supported by all co-authors." This appears to be inaccurate.***

The first author assumes responsibility for typographical and grammatical errors and we are now taking greater care in revising the English used.

Added Bibliography:

Carslaw, K. S., Wirth, M., Tsias, A., Luo, B. P., Dornbrack, A., Leutbecher, M., Volkert, H., Renger, W., Bacmeister, J. T., and Peter, T.: Particle microphysics and chemistry in remotely observed mountain polar stratospheric clouds, J. Geophys. Res., 103, 5785–5796, doi:10.1029/97JD03626, 1998.

Deshler, T., N. Larsen, C. Weisser, J. Schreiner, K. Mauersberger, F. Cairo, A. Adriani, G. Di Donfrancesco, J. Ovarlez H. Ovarlez, U. Blum, K.H. Fricke, and A. Dörnbrack, Large nitric acid particles at the top of an Arctic stratospheric cloud, *J. Geophys. Res.*, 108(D16), 4517, doi:10.1029/2003JD003479, 2003.

Engel, I., Luo, B. P., Pitts, M. C., Poole, L. R., Hoyle, C. R., Grooß, J.-U., Dörnbrack, A., and Peter, T.: Heterogeneous formation of polar stratospheric clouds – Part 2: Nucleation of ice on synoptic scales, Atmos. Chem. Phys., 13, 10769–10785, https://doi.org/10.5194/acp-13-10769-2013, 2013.

Grothe, H., Tizek, H., Waller, D., & Stokes, D. J. (2006). The crystallization kinetics and morphology of nitric acid trihydrate. Physical Chemistry Chemical Physics, 8, 2232–2239. https://doi.org/10.1039/B601514J

Hoyle, C. R., Engel, I., Luo, B. P., Pitts, M. C., Poole, L. R., Grooß, J.-U., and Peter, T.: Heterogeneous formation of polar stratospheric clouds – Part 1: Nucleation of nitric acid trihydrate (NAT), Atmos. Chem. Phys., 13, 9577–9595, https://doi.org/10.5194/acp-13-9577-2013, 2013.

Liu, L., and M.I. Mishchenko, 2001: Constraints on PSC particle microphysics derived from lidar observations. J. Quant. Spectrosc. Radiat. Transfer, 70, 817-831, doi:10.1016/S0022-4073(01)00048-6.

Molleker, S., Borrmann, S., Schlager, H., Luo, B., Frey, W., Klingebiel, M., et al. (2014). Microphysical properties of synoptic-scale polar stratospheric clouds: In situ measurements of unexpectedly large HNO3-containing particles in the Arctic vortex. Atmospheric Chemistry and Physics, 14(19), 10785–10801. https://doi.org/10.5194/acp-14-10785-2014

Snels, M., Cairo, F., Di Liberto, L., Scoccione, A., Bracaglia, M., & Deshler, T. (2021). Comparison of coincident optical particle counter and lidar measurements of polar stratospheric clouds above McMurdo (77.85°S, 166.67°E) from 1994 to 1999. Journal of Geophysical Research: Atmospheres, 126, e2020JD033572. https://doi.org/10.1029/2020JD033572

Woiwode, W., Grooß, J.-U., Oelhaf, H., Molleker, S., Borrmann, S., Ebersoldt, A., et al. (2014). Denitrification by large NAT particles: The impact of reduced settling velocities and hints on particle characteristics. Atmospheric Chemistry and Physics, 14(20), 11525–11544. https://doi.org/10.5194/acp-14-11525-2014

Woiwode, W., Höpfner, M., Bi, L., Khosrawi, F., & Santee, M. L. (2019). Vortex-wide detection of large aspherical NAT particles in the Arctic winter 2011/12 stratosphere. Geophysical Research Letters, 46, 13420–13429. https://doi.org/10.1029/2019GL084145

Woiwode, W., Höpfner, M., Bi, L., Pitts, M. C., Poole, L. R., Oelhaf, H., et al. (2016). Spectroscopic evidence of large aspherical β-NAT particles involved in denitrification in the December 2011 Arctic stratosphere. Atmospheric Chemistry and Physics, 16(14), 9505–9532. https://doi.org/10.5194/acp-16-9505-2016.